

# Mount Pinatubo's effect on the moisture-based drivers of plant productivity

Ram Singh[1,2], Kostas Tsigaridis[1,2], Diana Bull[3], Laura P Swiler[3], Benjamin M Wagman[3], Kate Marvel[2]

**Affiliations**

[1] *Center for Climate Systems Research, Columbia University, New York, USA*

[2] *NASA Goddard Institute for Space Studies, New York, NY-10025, USA*

[3] *Sandia National Laboratories, Albuquerque, NM, USA*

**Correspondence:** Ram Singh (rs4068@columbia.edu, ram.bhari85@gmail.com)

**Abstract**

Large volcanic eruptions can significantly affect the state of the climate, including stratospheric sulfate concentrations, surface and top-of-atmosphere radiative fluxes, stratospheric and surface temperature, and regional hydroclimate. The prevalence of higher natural variability in how the regional rainfall responds to the volcanic-induced climate perturbations creates a knowledge gap in understanding of how eruptions affect ecohydrological conditions and plant productivity. Here we will explore the understudied store (soil moisture) and flux (evapotranspiration) of water as the short-term ecohydrological control over plant productivity in response to the 1991 eruption of Mt. Pinatubo. We used the NASA's Earth system model for modeling of the 1991's Mt. Pinatubo eruption and detection of hydroclimate response. The model simulates a radiative perturbation of -5 Wm$^{-2}$ and mean surface cooling of ~ 0.5 °C following the Mt. Pinatubo eruption in 1991. The rainfall response is spatially heterogenous, due to dominating variability, yet still shows suppressed rainfall in the northern hemisphere after the eruption. We find that up to 10-15% of land regions show a statistically significant agricultural response. Results confirm that these higher-order impacts successfully present a more robust understanding of inferred plant productivity impacts. Our results also explain the geographical dependence of various contributing factors to the





compound response and their implications for exploring the climate impacts of such episodic
forcings.
**Introduction**

Volcanic eruptions are the most prominent source of sulfate aerosols in the stratosphere

and are among the natural drivers of climate variability. Volcanically-injected sulfate aerosols in
the stratosphere alter the Earth's radiative balance by simultaneously reflecting incoming solar
radiation and absorbing outgoing longwave radiation emitted from the Earth's surface (Robock,
2000). The presence of sulfate aerosol for months to years after an eruption and its microphysical
transformation in the stratosphere affect the climate system through numerous direct and indirect
effects  (Barnes and Hofmann, 1997; Brad Adams et al., 2003; Briffa et al., 1998; Deshler et al.,
2003; Lambert et al., 1993; LeGrande et al., 2016; LeGrande and Anchukaitis, 2015; Li et al.,
2013; Pinto et al., 1989; Santer et al., 2014; Sigl et al., 2015; Singh et al., 2023; Tejedor et al.,
2021; Toohey et al., 2019; Zambri et al., 2017; Zambri and Robock, 2016; Zhao et al., 1995).
The Mt. Pinatubo eruption (June 1991) remains the largest eruption in the satellite era, and it has
been explicitly documented and analyzed for its radiative and climate impacts. Numerous studies
based on satellite observations and supported through different modeling efforts estimated that
Mt. Pinatubo injected 10-20 Tg of $SO_2$ at a range of 18-25 km of plume height (Aquila et al.,
2012; Bluth et al., 1992; Dhomse et al., 2014; Gao et al., 2023; McGraw et al., 2024; Mills et al.,
2016; Sheng et al., 2015a, b; Stenchikov et al., 1998). The radiative impacts of Mt. Pinatubo's
eruption estimate an aerosol optical depth of 0.15 for 550 nm wavelength, with an effective
radius in the range of 0.16 to 1 micrometer (μm) and net radiative forcing on the order of 5-6
$Wm^{-2}$ (Lacis et al., 1992; Sato et al., 1993; Stenchikov et al., 1998). Estimates of the induced
surface cooling range between 0.3-0.7 °C; lower stratosphere warming estimates are in the range



of 2-3 °C (Bluth et al., 1992; Dutton and Christy, 1992; Hansen et al., 1992; Labitzke and
McCormick, 1992; Lacis et al., 1992; McCormick and Veiga, 1992; Minnis et al., 1993;
Ramachandran et al., 2000; Stenchikov et al., 1998 Hansen et al., 1993).

In this study, we aim to explore the mechanisms by which the Mt. Pinatubo eruption

affected the hydroclimatic conditions and water-based drivers of plant productivity. Agricultural
productivity is sensitive to changes in temperature and precipitation  (Lobell and Field, 2007;
Olesen and Bindi, 2002; Rosenzweig and Parry, 1994). Although much work has been devoted to
understanding Mt. Pinatubo's impacts on plant productivity, the literature has been dominated by
studies focusing on impacts from changes to the quantity and quality of incoming solar radiation
(Farquhar and Roderick, 2003; Gu et al., 2002, 2003; Jones and Cox, 2001; Robock, 2005).
Proctor et al. (2018) have estimated a decrease in C4 (maize) and C3 (soy, rice, and wheat)
agricultural crop production in response to Mt. Pinatubo driven mainly by changes in incoming
radiation. Krakauer and Randerson (2003) evaluated the role of surface cooling in reduced NPP
(Net Primary Productivity) using tree ring growth patterns following multiple Mt. Pinatubo-sized
eruptions in the last millennium. Reduced NPP was found in northern mid to high latitudes while
the signal in the lower latitudes and tropics was either not significant or constrained by the other
factors. Other studies have further expanded into societal impact research focusing on
volcanically induced poor harvest and agricultural productivity over different regions (Hao et al.,
2020; Huhtamaa and Helama, 2017; Manning et al., 2017; Singh et al., 2023; Toohey et al.,

2016).

Similarly, much work has been devoted to understanding the hydroclimate response to

Mt. Pinatubo through changes in atmospheric precipitation (Barnes et al., 2016; Paik et al., 2020;
Trenberth and Dai, 2007).  Monsoon seasonal rainfall decreases in the season after an eruption



(Colose et al., 2016; Iles et al., 2013; Liu et al., 2016; Singh et al., 2023; Tejedor et al., 2021).
However, rainfall alone provides an incomplete understanding of the drought conditions relevant
for plant productivity; a rainfall deficit could in principle be overcome by moisture stored in soil.
Hence, meteorological drought indices (e.g. SPI (McKee et al., 1993)  or PDSI (Palmer 1965))
based on rainfall ignore a full water balance approach. Furthermore, meteorological drought
indices tend to be designed to evaluate prolonged periods of abnormally dry weather conditions.
For instance, PDSI is an indicator of drought with a 9-month horizon (Mullapudi et al., 2023).
Yet, agricultural crops are heterogeneously sensitive to the timing and degree of moisture deficits
during particular portions of the crop growth cycle; for instance, corn yield can be decreased by
as much as 25% for a 10% water deficit during the pollination stage (Hane and Pumphrey, 1984).
Thus, consideration of indices with high temporal frequency can be especially important when
focusing on agriculture.

Soil moisture is the stock of water stored underground and is a primary source for the

flux of water to the atmosphere and plants through evapotranspiration. Energetically, evaporation
of water from bare surface soil or transpiration of water during photosynthesis in plants from the
root zone soils is demanding, using a dominant portion of absorbed solar energy (Trenberth et al.,
2009). Plant transpiration is the largest contributor to land evapotranspiration (Nilson and
Assmann, 2007; Seneviratne et al., 2010 and references therein). Soil moisture decrease in the
root zone establishes an important control over plant productivity as transpiration is an integral
component of photosynthesis (Chen and Coughenour, 2004; Denissen et al., 2022). Multiple
studies have established that water supply is the limiting factor for climatic evapotranspiration
over tropical and subtropical land areas while temperature is an important controlling factor in
northern mid- and high latitudes (Dong and Dai, 2017 and references therein). However, soil



moisture changes in response to Mt. Pinatubo eruption (1991) are largely underreported in the
literature and it is unclear how soil moisture would respond given volcanically forced changes in
primary drivers (temperature and precipitation).

To our knowledge, no study has yet investigated multiple indicators of water use with

agricultural productivity after a short-duration event like Mt. Pinatubo eruption. Hence, this
study looks to explicitly investigate changes in agricultural drought indices from Mt. Pinatubo by
considering the store (soil moisture) and flux (evapotranspiration) of water as potential short-
term controls over productivity in particular regions. We use NASA's state-of-the-art Earth
system model with interactive aerosol chemistry to conduct the simulation experiments
consistent with the counterfactual inference of causation approach for the Mt. Pinatubo eruption.
The Mt. Pinatubo effect in the model-simulated climate is evaluated through the various
pathways of climate impacts, from the primary dependent variables to the higher order responses
controlling plant productivity. Considering the complexity of modeling the terrestrial system,
vegetation demographics, and physiological characteristics, we use the soil moisture and
evapotranspiration-based agricultural drought indices SMDI (soil moisture deficit index) and
ETDI (evapotranspiration deficit index) developed by (Narasimhan and Srinivasan, 2005) to
account for agricultural productivity. We evaluate short-term (weekly) and long-term (seasonal)
scale changes in SMDI and ETDI relative to statistics over longer modern time-period. By
focusing on soil moisture and evapotranspiration metrics, the major water-based drivers of plant
productivity are explored to deepen our understanding of the impacts Mt. Pinatubo had on plant
productivity.
**2.0 Method, Experiment, and Data**



**2.1 NASA GISS ModelE2.1 (MATRIX)**: We use the state-of-the-art Earth system model from
the NASA (National Aeronautics and Space Administration) Goddard Institute for Space Studies,
NASA GISS ModelE2.1 (Bauer et al., 2020; Kelley et al., 2020). NASA GISS ModelE2.1 has an
atmospheric horizontal latitude-longitude grid spacing of 2.0x2.5 degrees (at the equator) with 40
vertical levels and a model top of 0.1 hPa. We used the interactive chemistry version MATRIX
(Multiconfiguration Aerosol TRacker of mIXing state) aerosol microphysics module  (Bauer et
al., 2008, 2020), which is based on the Quadrature Method of Moment (QMOM) to predict
aerosol particle number, mass, and size distribution for 16 different mixed modes of the aerosol
population. New particle formation is represented by Vehkamäki et al. (2002), along with
aerosol-phase chemistry, condensational growth, coagulation, and mixing states (Bauer et al.,
2013). 16 mixing states with 51 aerosol tracers for sulfate, nitrate, ammonium, aerosol water,
black carbon, organic carbon, sea salt, and mineral dust are resolved in this microphysical
module (Bauer et al., 2008, 2020). The first indirect effect of aerosols in terms of changes in
cloud properties through nucleation is also computed within MATRIX.

The ocean component (GISS Ocean v1) of the model has a horizontal resolution of

1x1.25 degrees, with 40 vertical layers. The land component is the Ent Terrestrial Biosphere
Model (TBM) (Kim et al., 2015; Kiang 2012) which includes an interactive carbon cycle (Ito et
al., 2020), satellite-derived (MODIS) plant functional type and monthly variation of leaf area
index (Gao et al., 2008; Myneni et al., 2002), and tree height (Simard et al., 2011). Interannual
variations in the vegetation properties are controlled by rescaling the vegetation fraction (Figure
S6) using historical crops and pasture at grid scale to account for land cover and land use
changes (Ito et al., 2020; Miller et al., 2020). The land model has two defined tiles for the soil
layer: bare and vegetated, and each has six vertical levels to a depth of 3.5 m (11.5 feet)



(Rosenzweig and Abramopoulos, 1997). Rooting depth for different plant functional type are
also given by Rosenzweig and Abramopoulos (1997) and more than 60% of roots for crop plant
functional type are located within 0.6 m (1.96 feet) of soil depth. In this version of the model, for
the agricultural grid cells, crop plant functional type and crop calendar are prescribed according
to McDermid et al., (2019). Irrigation in the GISS ModelE is implemented using the water
irrigation demand data (IWD; (Wisser et al., 2010) and irrigation potential calculations based on
(Wada et al., 2013) as discussed in (Cook et al., 2020).
**2.2 Experiment Design**: The MATRIX version of GISS ModelE2.1 with active tracers is three
times more computationally expensive than the non-interactive (prescribed pre-calculated
aerosol concentration and extinction) version. We extended an equilibrated 1400-yearlong PI
control run with non-interactive tracers with an additional 500 years using the MATRIX version
with prognostic tracers before starting the 'historical' run. MATRIX includes the tropospheric
chemistry scheme that includes the inorganic (Ox NOx Hox and CO), organic chemistry of CH4,
and higher hydrocarbons (Gery et al., 1989; Shindell, 2001; Shindell et al., 2003). The
stratospheric chemistry includes bromine, chlorine, and polar stratospheric clouds (Shindell et
al., 2006). Dust emission in the model is controlled by the climate variables such as winds and
soil moisture at the spatial and temporal scales (Miller et al., 2006). However, anthropogenic
dust is not included in GISS ModelE2.1. Other anthropogenic emissions, including biomass
burning (pre-1997 from (van Marle et al., 2017) and 1997 onwards from the GFED4s inventory
(van der Werf et al., 2017)), are taken from the Community Emission Data System (CEDS)
inventory (Hoesly et al., 2018). Most importantly, the volcanic $SO_2$ forcing for the 'historical'
run (1850-1977) is the daily emission rate from VolcanEESM (Neely and Schmidt 2016 :
https://catalogue.ceda.ac.uk/uuid/a8a7e52b299a46c9b09d8e56b283d385 ) and satellite



measurement driven $SO_2$ inventory (Carn et al., 2017) for 1978 to 2022. The cumulative Mt.
Pinatubo emission is 15194 kt (~15.2 Tg) of $SO_2$ injected from 12th to 16th of June 1991 above
the Mt. Pinatubo vent, with a maximum of 15000 kt (15 Tg) emitted on June 15th at a plume
height of 25 km (Diehl et al., 2012). The MATRIX version of the GISS ModelE2.1 used for all
of our simulations predicts the nucleation, evolution, and removal of sulfate aerosols
prognostically.
Table 1: Simulation experiment design.

| EXP Name | Description | Time period /run length | Ensembles | Configuration |
|---|---|---|---|---|
| GISS-CMIP6-PI | Preindustrial | 1850 climatology /1300 years | 1 | GISS ModelE2.1 – MATRIX with prescribed stratospheric aerosols (average volcanic AOD for historical period, 1850-2014) |
| GISS-PI | Preindustrial | 1850 climatology /500 years | 1 | Extension to GISS-CMIP6-PI using GISS ModelE2.1 – MATRIX with prognostic tracers |
| GISS-HIST-SO$_2$ | historical | 1850-2014 /165 years | 1 | GISS ModelE2.1 – MATRIX with all forcings as specified by CMIP6 except daily emission rate of injection of $SO_2$ (VolcanEESM; Neely & Schmidt 2016; Carn et al., 2017) |
| GISS-PIN-SO$_2$ | historical | 1986-1999 / 15 years | 11* | GISS ModelE2.1 – MATRIX Branched out from GISS-HIST-SO2, with all forcings as specified by CMIP6 except daily emission rate of $SO_2$ from a combination of VolcanEESM (Neely & Schmidt 2016) and Carn et al., 2017. |
| GISS-NOPIN-SO$_2$ | historical | 1986-1999 / 15 years | 11* | GISS ModelE2.1 – MATRIX with all forcings as according to CMIP6 with daily emission rate |



| | | | | of SO$_2$ **without Mt. Pinatubo** from a combination of VolcanEESM (Neely & Schmidt 2016) and Carn et al., 2017. |
|---|---|---|---|---|

\*These ensemble members are branched out from the GISS-HIST-SO$_2$ by perturbing a
radiation-related random number generator that deals with fractional cloudiness in the
column.

**2.3 Methods:** This study aims to explore the impacts of the Mt. Pinatubo eruption on the major
drivers of primary productivity, focusing on soil moisture-related metrics and evapotranspiration.
Hereafter, we use the terminology 'PCH' (Mt. Pinatubo and Cerro Hudson) to refer to the 'GISS-
PIN-SO2' and 'NP' for the counter-factual ensemble 'GISS-NOPIN-SO2'. Since we are focusing
on the Mt. Pinatubo driven climate response, we have included the Cerro Hudson eruption in
both ensembles.
**2.3.1 Statistical analysis used to detect Mt. Pinatubo-significant regions and calculate their**
**anomalies.**
We treat the no-Pinatubo ensemble (NP) as a counter-factual climate simulation and utilize it to
perform the paired Student's t-test. The null hypothesis is that the ensemble means of a quantity
of interest (QoI) in a region over a time period are the same between ensembles: $H_o: \bar{\mu}_{PCH} =$
$\bar{\mu}_{NP}$. Regions filled in grey in subsequent figures in this document indicate that the null
hypothesis cannot be rejected at the 95% confidence level.  Regions in which the null hypothesis
is not accepted are highlighted in color in subsequent plots, where the color hues show anomalies
with respect to the simulated historical climatology for the period 1950-2014.  The coloring is
done to emphasize significant regions of anomalies, but we emphasize the difference in
calculations: the grey regions show no significant change between the PCH and NP ensembles,
while the anomalies are PCH ensemble mean minus climatology.
**2.4 Impact metrics**



The distribution of incoming and outgoing radiation influences the hydrological cycle (Kiehl and
Trenberth, 1997; Trenberth and Dai, 2007). A reduction of solar radiation at the surface has the
potential to reduce rainfall and change the latent heat-dominated atmospheric heating pattern
(Trenberth and Stepaniak, 2004). The perturbed atmospheric conditions and surface energy
budget could affect soil moisture. Along with the surface air temperature and precipitation, we
use soil moisture and surface energy budget-oriented drought indices (the soil moisture deficit
index (SMDI) and evapotranspiration deficit index (ETDI)) to evaluate the land-atmosphere
interaction and account for the potential drivers to the crop plant productivity in the model
simulated post- Mt. Pinatubo environmental conditions (Narasimhan and Srinivasan 2005).
SMDI represents the land-based soil moisture state in selected depth horizons (i.e. SMDI_2
means Soil Moisture Deficit Index for 2 feet (0.6 m) depth). ETDI represents the atmospheric
conditions governing the land-atmosphere interaction and is an indicator of plant health. Lastly,
plant transpiration is analyzed to the explore the simulated physiological response to the
volcanically induced hydroclimatic conditions.

The Palmer drought severity index (PDSI) and other indices are commonly used to

represent climatological drought conditions, but we focus on SMDI and ETDI because these can
represent short-term developing agricultural drought conditions as a response to plant
productivity and are free from the limitations of other metrics like PDSI. For example, SMDI
and ETDI are seasonally independent measures and are comparable across space, even for
different climatic zones.

SMDI and ETDI were calculated as described in Narasimhan and Srinivasan (2005) using

model output at monthly and daily scales. Daily model output is resampled weekly to compute
the indices. Weekly frequency is used because it is suitable for agricultural applications and the





daily frequency is comparatively higher and computationally expensive for such indices. Below
we reproduce the weekly calculation of SMDI and ETDI as presented in Narasimhan and
Srinivasan (2005).
**2.4.1 Soil Moisture Deficit Indices (SMDI)** : Soil moisture deficit index measures the
wetness/dryness of the soil moisture condition in comparison to long term records spanning

1950-2014.

$SD_{i,j} = \frac{SW_{i,j} - MSW_j}{MSW_j - minSW_j} \times 100$  if $SW_{i,j} \leq MSW_j$        ….. (2.4.1a)
And
$SD_{i,j} = \frac{SW_{i,j} - MSW_j}{maxSW_j - MSW_j} \times 100$  if $SW_{i,j} > MSW_j$      ….. (2.4.1b)
$SD_{i,j}$ is the soil water deficit (%) for week j of year i. $SW_{i,j}$ is the mean weekly soil water
available in the soil profile (mm) for week j of year i, $MSW_j$ is the long-term (calibration period)
median available water in the soil profile (mm) for week j, and $minSW_j$ and $maxSW_j$ are the $j^{th}$
weekly minimum and maximum of soil water available in the soil profile across the calibration
period (1950-2014). The soil mositure deficit index for any given week can be calculated as
$SMDI_j = 0.5 * SMDI_{j-1} + SD_j/50$        ….. (2.4.1c)
SMDI can be calculated for different depths of soil; we used the 2, 4 and 6 feet depths for SMDI
estimation, approximately 0.6, 1.2, and 1.8 meters, respectively. For SMDI, it is typical to use
feet instead of meters in the literature, which is why we use the same units here. SMDI-4 means
we considered the soil moisture content between 2 to 4 feet depth. Similarly, SMDI-6 indicates
the soil moisture content between 4 to 6 feet in depth.





**2.4.2 Evapotranspiration Deficit Index (ETDI):** The limitations of the Palmer Drought

Severity Index (PDSI; Palmer, 1965) and Crop Moisture Index (CMI; (Palmer, 1968)) in the

formulation used for PET calculation Thornthwaite, (1948) and lack of accountability to the land

cover type on water balance encouraged the exploration of ETDI for agricultural productivity.

Also, in the climate models, surface energy fluxes are parameterized in terms of the

thermodynamical gradient of atmosphere and land models and thus represent the land-

atmosphere interactions not accounted for by these atmosphere-only indices. We utilized model

simulated surface energy fluxes (Latent and Sensible heat) to calculate the potential (PET) and

actual evapotranspiration (AET) to estimate the water stress ratio. However, the applicability of

the Penman-Monteith equation for reference crops Allen et al., (1998) provides a substitute

method for PET calculation, which, although not shown, broadly produced similar results.

In Equation 2.4.2a and 2.4.2b  we used the model simulated energy fluxes to calculate AET and

PET as suggested in (Milly and Dunne, 2016; Scheff and Frierson, 2015).

The energy budget equation at the surface is given by Rn= G + LH+SH, where Rn is

incoming solar radiation, G is ground energy, LH and SH represent the Latent and Sensible heat

fluxes, respectively. We then use these to calculate PET and AET (unit as mm per day):

$$PET = 0.8(R_n - G) = (0.8 * 0.0864/2.45) * (LH + SH)$$   ……. (2.4.2a)

And

$$AET = LH* (0.0864/2.45).$$                   ……. (2.4.2b)

The evapotranspiration deficit index is estimated using the water stress condition using the actual

evapotranspiration (AET) and potential evapotranspiration (PET) per grid cell as given below.

$$WS = \frac{PET-AET}{PET}$$      ….. (2.4.2c)





WS ranges between 0 to 1, where 0 signifies that evapotranspiration is happening at potential
rate and 1 stands for no actual evapotranspiration. WS represents the water stress ratio at a
monthly or weekly basis ($WS_j$), which is further utilized to calculate water stress anomaly
($WSA_{i,j}$) for week j of year i as given below.
$WSA_{i,j} = \frac{MWS_j - WS_{i,j}}{MWS_j - minWS_j} \times 100$  if $WS_{i,j} \leq MWS_j$  …… (2.4.2d)
And
$WSA_{i,j} = \frac{MWS_j - WS_{i,j}}{maxWS_j - MWS_j} \times 100$  if $WS_{i,j} > MWS_j$   ……. (2.4.2e)
Here, $MWS_j$, $minWS_j$, and $maxWS_j$ represent the longterm median, minimum, and maximum of
the water stress ratio over the calibration period. Water stress anomaly ranges between -100% to
100%, indicating very dry to wet conditions over the region.
Finally the severity of the drought condition is calculated as ETDI, similar to SMDI (equation
2.4.1c) at a monthly/weekly time scale.
$ETDI_j = 0.5 * ETDI_{j-1} + WSA_j / 50.$      ……... (2.4.2f)
The indices SMDI and ETDI range from -4 to +4, representing the excessive wet and dry
conditions. The bounding values -4 or +4 represent extremely dry/wet conditions as the
deficit/excess of soil-moisture deficit (SM) or water stress anomaly (WSA) is reached, relative to
the maximum over the reference calibration period.
We also highlight the justification for selecting 1950-2014 as the base period for
analyzing the response in climate variables and the long-term calibration period for drought
indices calculations. (**Supplementary information** section 1s).
**3.0 Results**
The result section of this study first presents the NASA GISS model's simulated
properties of the 1991 Mt. Pinatubo eruption and then further evaluates the primary (aerosol



optical depth, radiation, and temperature) and secondary (precipitation, soil moisture,
evapotranspiration, and transpiration) impacts on plant productivity.
**3.1 Radiative forcings and response**

We analyze the microphysical and radiative properties of volcanic aerosol simulated by

the NASA GISS ModelE (MATRIX) in the PCH ensemble set. The current setup of GISS
ModelE uses the aerosol microphysical module MATRIX represent the various states and
provide particle number, mass, and size information for different mixed modes of the aerosol
population. In the simulation of the Mt. Pinatubo eruption, the volcanically injected $SO_2$ in the
stratosphere oxidizes in the presence of prognostically evolving OH radicals to form the
stratospheric sulfate aerosols. Sulfate aerosols grow by condensation of gas (nucleation, and self-
coagulation (preexisting)) to the Aitken (AKK) mode (mean mass diameter <0.1 μm), and further
growth in size leads to the transfer to Accumulation (ACC) mode (Bauer et al., 2008; Bekki,
1995). The transfer between the two particle modes is controlled through the transfer function
based on particle mean mass diameter (Bauer et al., 2008). GISS ModelE (MATRIX) PCH
simulated a sulfate aerosol size with an effective radius ($R_{eff}$ ) of the order of 0.3-0.6 μm after the
Mt. Pinatubo eruption (not shown), consistent with several observation and modeling estimates
(Bauman et al., 2003; Bingen et al., 2004; Russell et al., 1996; Stenchikov et al., 1998). GISS
ModelE (MATRIX) PCH simulated a peak global mean aerosol optical depth (AOD; for 550 nm
wavelength) of 0.21 (Supplementary Fig S2 bottom panel) a few months after the eruption,
which then decreases due to deposition (English et al., 2013; Sato et al., 1993). Here, the model-
simulated extinction of the radiation (AOD) due to volcanic aerosol and radiative forcing is
larger than previously reported AOD of 0.15 and forcing of -4.0 to -5.0  Wm$^{-2}$ due to the Mt.
Pinatubo eruption (Hansen et al., 1992; Lacis et al., 1992).



The mass and size of volcanic sulfate aerosol firmly control the scattering of the
incoming shortwave radiation and the absorption of longwave (Brown et al., 2024; Kinne et al.,
1992; Lacis, 2015; Lacis et al., 1992; Lacis and Hansen, 1974). The first-order climate response
to the volcanically-injected sulfate aerosol in the stratosphere is the perturbation of the radiative
balance of the Earth system (Hansen et al., 1980; Lacis et al., 1992; Stenchikov et al., 1998).
Figure 1 shows that the GISS ModelE PCH has simulated a peak longwave, shortwave, and net
radiative response of +3.0 $Wm^{-2}$, -8.0 $Wm^{-2}$, and -5.0 $Wm^{-2}$ respectively, a few months after the
eruption, which recovers slowly in next 24 months and is consistent with previous studies
(Stenchikov et al., 1998; Hansen et al., 1992; Minnis et al., 1993; Brown et al 2024). These
radiative responses are calculated with respect to the climatology for the period 1950-2014 in
GISS-Hist-SO2.  The GISS model also simulated a smaller peak ranging within 1 $Wm^{-2}$ in the
counterfactual (without Mt. Pinatubo) runs, likely due to the Cerro Hudson eruption in August

1991.

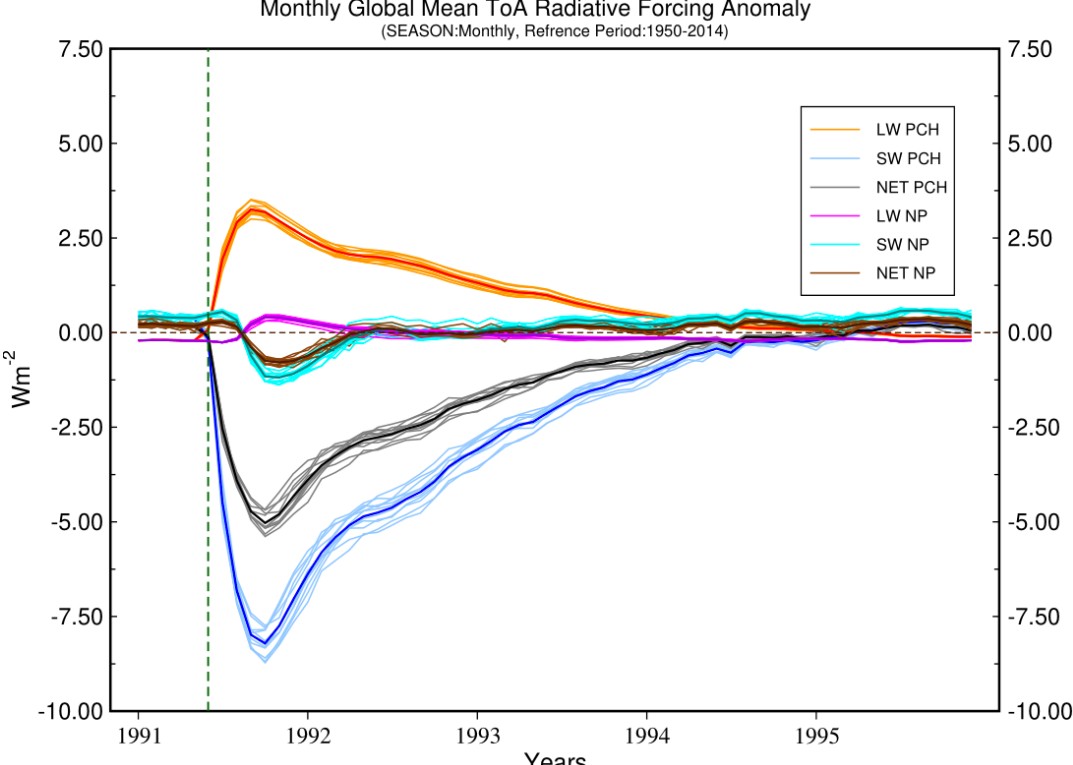

Figure 1. Monthly anomaly of longwave, shortwave, and net radiative forcing simulated by the GISS ModelE for Mt. Pinatubo (PCH) and counterfactual (NP) ensemble. The response anomalies are calculated with respect the climatology for the period 1950-2014, taken from the GISS historical runs (GISS-HIST-SO2). The light-colored thin lines represent the individual ensemble member, and the dark broad line is multi-ensemble mean for each variable (longwave, shortwave and net radiative response).

**3.2 Aerosol dispersion and Temperature Response**

Figure 2 shows the zonal mean anomaly for the aerosol optical depth (AOD), lower stratosphere temperature (MSU-TLS satellite simulator), and surface temperature. The zonal AOD shows the dispersion and transport of aerosol poleward after the eruption. Horizontal dispersion and transport of the aerosols is strictly influenced by the stratospheric meteorology and atmospheric circulation, which is independent in each ensemble member, and depends on the





plume height and season. GISS ModelE has simulated AOD consistently with previous studies
(Aquila et al., 2012; Brown et al., 2024; Rogers et al., 1998; Timmreck et al., 1999; Trepte et al.,
1993). Cross-equatorial dispersion to the southern hemisphere might be due to the more robust
Brewer-Dobson circulation in the austral winter (Aquila et al., 2012). Meanwhile, the phases of
QBO and local heating also play a crucial role in the poleward and vertical dispersion of
stratospheric aerosols (Hitchman et al., 1994; Ehrmann et al., 2024 (in-prep)). A smaller peak in
the southern hemisphere (45° S) in later 1991 likely due to the Cerro Hudson eruption, which
injected ~1.5 Tg of $SO_2$ at a height of 15 km.

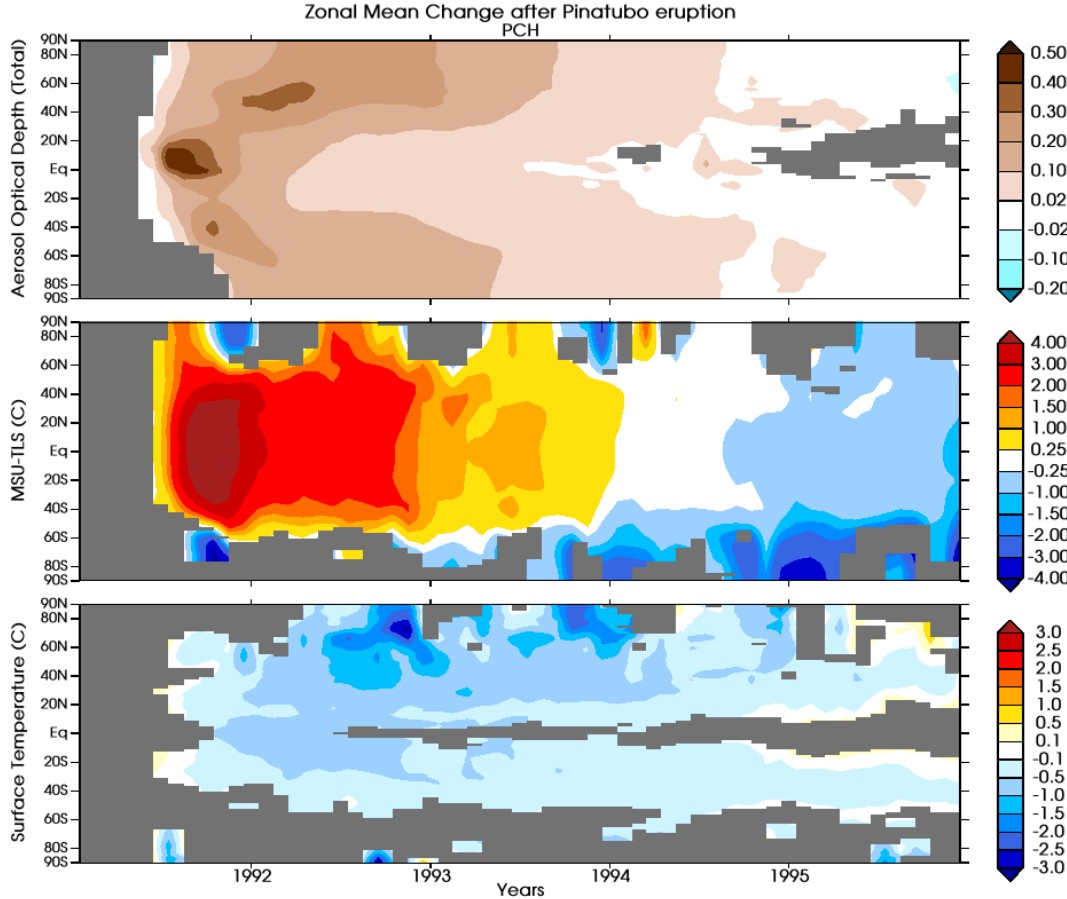




Figure 2. Zonal mean of monthly anomalies for multi ensemble mean for aerosol optical depth at
550 nm (top panel), microwave sounding unit temperature (MSU-TLS) for lower stratosphere
(middle panel) and surface air temperature with respect to the 1950-2014 climatology. Gray
regions show no statistically significant difference between the PCH and NP response. The
colored areas show anomalies of PCH with respect to the climatology from 1950-2014.

The middle and lower panels in Figure 2 show the model-simulated microwave sounding unit
(MSU) temperature for the tropical lower stratosphere (TLS) response due to absorption of
longwave radiation and for the surface temperature response due to the net radiative perturbation,
which is dominated by the scattering of incoming solar radiation. The model simulates a peak
warming of over 4 °C in the tropical lower stratosphere shortly after the eruption, which lasts for
a few months when the concentration of sulfate aerosols is highest. Significant warming in the
range 2-3 °C lasts until the end of 1992, and overall simulated stratospheric warming is
consistent with previous studies. Figure S2 (top panel) shows a steplike transition with time with
a global mean increase of 3.0 °C in the lower stratosphere temperature after the Mt. Pinatubo
eruption followed by a trend consistent with Ramaswamy et al., (2006). The zonal structure of
surface temperature shows that the surface cooling follows the aerosol optical depth pattern, and
the greatest cooling is simulated in high latitudes. Temporal characteristics of lower stratosphere
warming, and surface cooling also show the seasonal variations of sunlight in northern polar
latitudes.
The spatial pattern of surface air temperature response is evaluated at the seasonal scale
for each year from 1991 to 1995 as shown in Figure 3. We conclude that the volcanic forcing
from the Mt. Pinatubo eruption results in a statistically different seasonal mean surface air
temperature response. Figure 3 shows that a spatial pattern of surface cooling starts appearing
after a few months of the eruption (during the SON season of the year 1991) when the gaseous



SO$_2$ is oxidized into sulfate aerosols. The surface cooling signature due to the volcanic aerosols
is significant in 1992 and 1993 before recovering in 1994 towards pre-eruption temperature
conditions. The highest surface cooling is noticed over the sub-tropics and higher latitude land
regions in the northern hemisphere and reaches up to 2.5 °C at a regional scale.

To summarize:  the PCH GISS ModelE simulated global mean peak cooling response is

~0.5 °C after the eruption as shown in Supplement figure S2 (Middle panel) with a range
between 0.25 – 1.0 °C for individual ensemble members, and this is consistent with the various
observation and modeling studies (Brown et al., 2024; Dutton and Christy, 1992; Hansen et al.,
1996; Minnis et al., 1993; Parker et al., 1996; Ramachandran et al., 2000; Stenchikov et al.,
1998)( Kirchner et al., 1999).



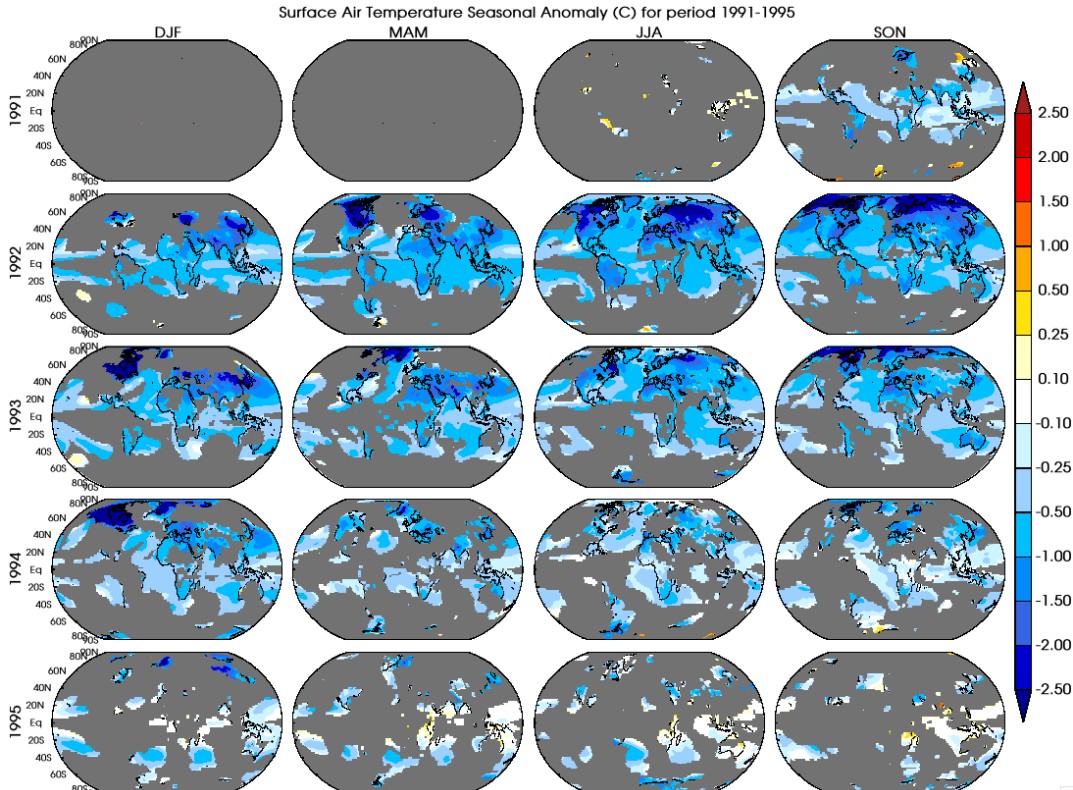

Figure 3: Seasonal mean surface temperature anomalies (°C) from the year 1991 to 1995 with

respect to the reference period of 1950-2014.  A grey color is painted over the grid cells where

the surface temperature anomalies are not statistically significant in comparison to the counter-

factual ensemble. The colored areas show anomalies of PCH with respect to the climatology

from 1950-2014.

**3.3 Rainfall Response**

Precipitation, presented seasonally for year of eruption (1991) and following year (1992)

in Figure 4, shows a highly complex and variable response to the volcanically induced

tropospheric cooling and radiative balance perturbation because of its sensitivity to the other

climate system components. Studies have shown that global mean precipitation decreases after

large volcanic eruptions (Gu et al., 2007; Gu and Adler, 2012; Iles et al., 2013; Robock and Liu,





1994; Singh et al., 2023; Trenberth and Dai, 2007). Colose et al., (2016) have postulated that the
asymmetrical surface cooling and radiative balance perturbation create an energetic deficit in the
hemisphere of eruption that constrains the poleward propagation of tropical rainfall belt (ITCZ)
in that hemisphere. In the case of the Mt. Pinatubo eruption, the PCH simulations show that
regional patches of significant decrease of up to 1 mm per day are spotted over tropical and
northern hemispheres (Africa, eastern and northern Asia) after the eruption (Figure 4). Also,
increasing rainfall patterns are simulated over the Mediterranean and European regions. Broadly,
the confidence level of precipitation response due to volcanic aerosols is strongly influenced by
the uncertainty due to many possible factors and prominent modes of atmospheric variability,
such as the strength of El Nino (Paik et al., 2020).

The zonal mean of the rainfall response (Figure S3) shows a clear decreasing trend in the

northern hemisphere tropical and higher latitudes with a positive rainfall response band around
20° N.  The PCH modelled rainfall response due to the Mt. Pinatubo eruption is broadly
consistent with the previous studies (Joseph and Zeng, 2011; Liu et al., 2016; Trenberth and Dai,
2007), but the uncertainty in rainfall response is still high. Although we use statistical
significance as our metric of determining significant anomalies, we do not deny erroneous
signals due to the model's internal variability when averaging the impacts across multiple
ensembles (Polvani et al., 2019). The inconsistency and complexity in the precipitation response
drives us to explore the compound hydroclimatic pathways of impacts beyond the rainfall such
as droughts.

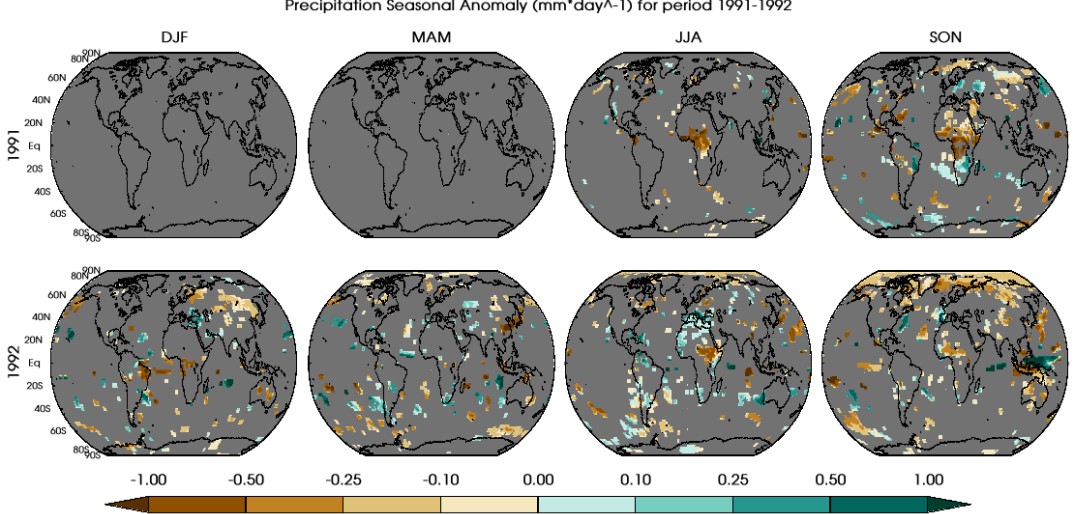

Figure 4. Seasonal mean precipitation anomalies (mm per day) from the year 1991 and 1992 with

respect to the reference period of 1950-2014. A grey color is painted over the grid cells where the

precipitation anomalies are not statistically significant in comparison to the counter-factual

ensemble. The colored areas show anomalies of PCH with respect to the climatology from 1950-

2014.

**3.4 Drought Conditions**

Land-atmosphere interaction under a radiatively perturbed environment plays a crucial

role in regulating the climate response at regional and sub-regional scales.  Changes in land-

atmosphere interactions on short timescales can strongly affect plant productivity.   Even short-

lived adverse conditions in the growth cycle have the potential for outsized impacts, especially if

they happen at a particular time in the growing cycle. Hence, we explore the weekly aspects of

these drought conditions in Section 4 to explore the temporal characteristics of variability in the

conditions.

**3.4.1 Seasonal Soil Moisture Drought Index (SMDI)**

The root zone is commonly defined as the top 3 – 6 feet of the soil column (Keshavarz et

al., 2014, and references therein) but most agricultural crops have shallower root systems
confined to the top 2 feet (Narasimhan and Srinivasan 2005). Hence, we focus on the soil
moisture deficit index (SMDI) (Narasimhan and Srinivasan 2005) for the top 2 feet of ground
depth (SMDI_2) as shown in Figure 5. As anticipated, more land area is covered by statistically
different SMDI_2 then in Figure 4 helping to further our analysis of water-driven impact to plant
productivity more then with precipitation.

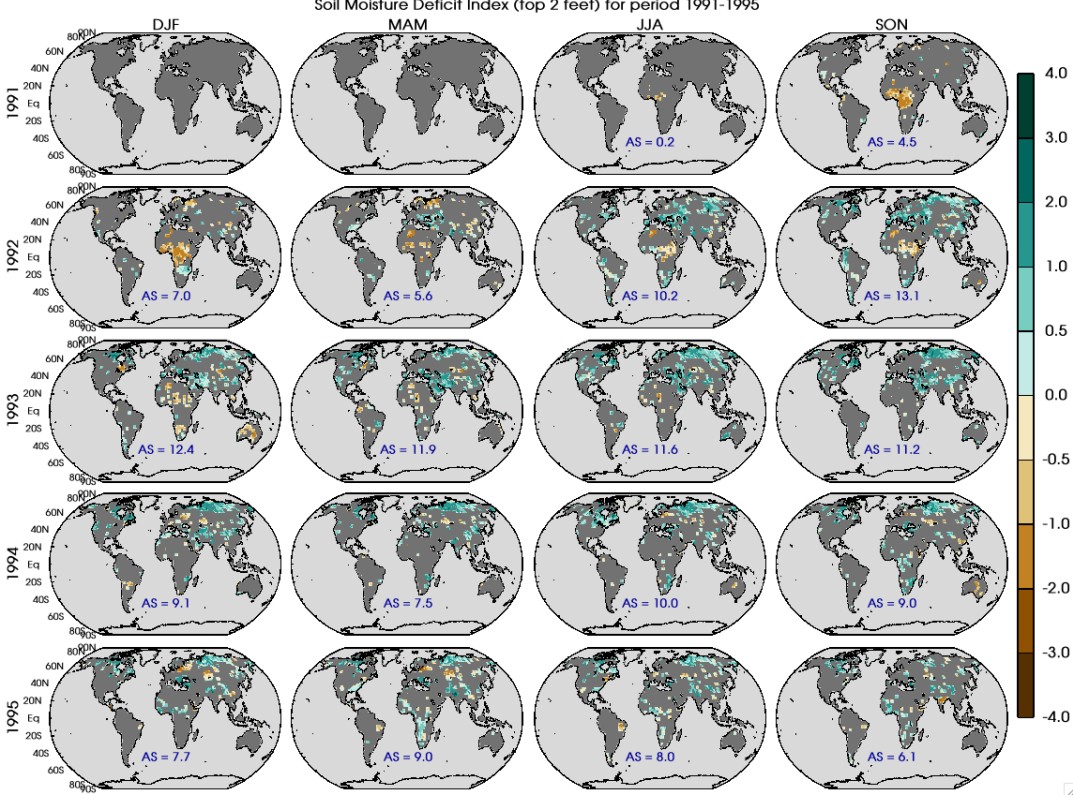


Figure 5. Soil moisture deficit index (SMDI_2) for the top 2 feet of ground depth evaluated
seasonally from 1991 to 1995. Grey color is painted over the grid cell where the SMDI_2 is not
statistically significant in contrast to counter-factual ensemble. The parameter AS on each panel



mark the percentage of land area which has shown statistically significant dry or wet response
after Mt. Pinatubo eruption.

Figure 5 clearly shows that the equatorial region, especially over Africa, has a significant

drying response due to Mt. Pinatubo in comparison to long term historical data starting from the
SON season of 1991 through the following DJF season. Although less robust, the dryness in this
region lasted through MAM of 1993. Severity of drying response reaches up to -2.0 on a scale of
extreme wet/dry at 4.0/-4.0, where a severity of – 4.0 reflects the maximum dryness (rarest case)
over the entire 1950-2014 calibration period. Figure 4 shows a similar pattern in equatorial
African rainfall decrease. Decrease in rainfall was present the first season post Mt. Pinatubo
eruption indicating an expected lagged response in SMDI_2.  Spatial coherence between these
signals is again re-established in the JJA and SON 1992 seasons albeit with more variation in
strength of signal.

Meanwhile, in the high latitudes of the northern hemisphere, we see an increase in the

store of soil moisture despite a decrease in rainfall in higher latitudes.  An exception to this is the
Mediterranean (extending towards east Mediterranean and western Asian) region where soil
moisture and rainfall, both show an increase during post- Mt. Pinatubo period. This increase in
the soil moisture in northern hemisphere is comparatively more pronounced in summer months
in comparison to the winter seasons. Thus, despite less water supply through rainfall, there is
persistent increase in soil-moisture in the root zone layer starting from JJA season of 1992. This
is likely due to less water extracted from this layer through evaporation and transpiration as well
as the implemented irrigation in GISS modelE (details in further sections).

Overall, Figure 5 shows equatorial drying signals mostly dominated through the DJF

season of the year 1993, but the wet conditions over higher latitudes lasted till 1995. Broadly, 6-



13% of the land region has shown a statistically significance response in terms of dry/wet
condition by the end of year 1995 because of the Mt. Pinatubo forcing.

Deeper soil layers approximate longer-term meteorologically defined drought indices

better (Narasimhan and Srinivasan 2005). This makes intuitive sense: precipitation provides the
recharge for the store of soil moisture and if there is a longer-term decline in precipitation all
hydraulically available moisture will be used for plant transpiration (both the deeper stores of
water and the soil-penetrating precipitation available) not allowing for deeper depth recharge.
Here we evaluate discrete layer depths instead of cumulative depths for two reasons. First, the
soil permeability changes with the depth and the inclusion of top layers erroneously reflects the
SMDI_2 signal in potentially impermeable regions; second, the SMDI_2 signal gets superposed
over the deeper layer response and misleads the actual soil moisture response for the deeper
layers.

As expected, when we evaluate the soil moisture deficit response between 2-4 feet soil

depth (SMDI_4) in Figure 6, and 4-6-feet soil depth (SMDI_6) in Figure S4, we see similar
spatial and temporal distributions as shown in Figure 5 with a corresponding decrease in the
percentage of area response. Spatially we see high latitudes across North America and
Northeastern and western Asia, equatorial Africa, European, and Mediterranean regions maintain
their SMDI-2 trend in Figure 5. However, the total area of response decreases from peak
coverages of 12-13% in SMDI-2 to less than 10-12% in SMDI-4 and 7-10% in SMDI_6 (shown
in Figure S4). Additional decreases in the degree of impacts are also seen between the three soil
layers. Note that the light grey colored regions in Figures 6 and S4 represent regions of
impermeability which does affect the total area of response.

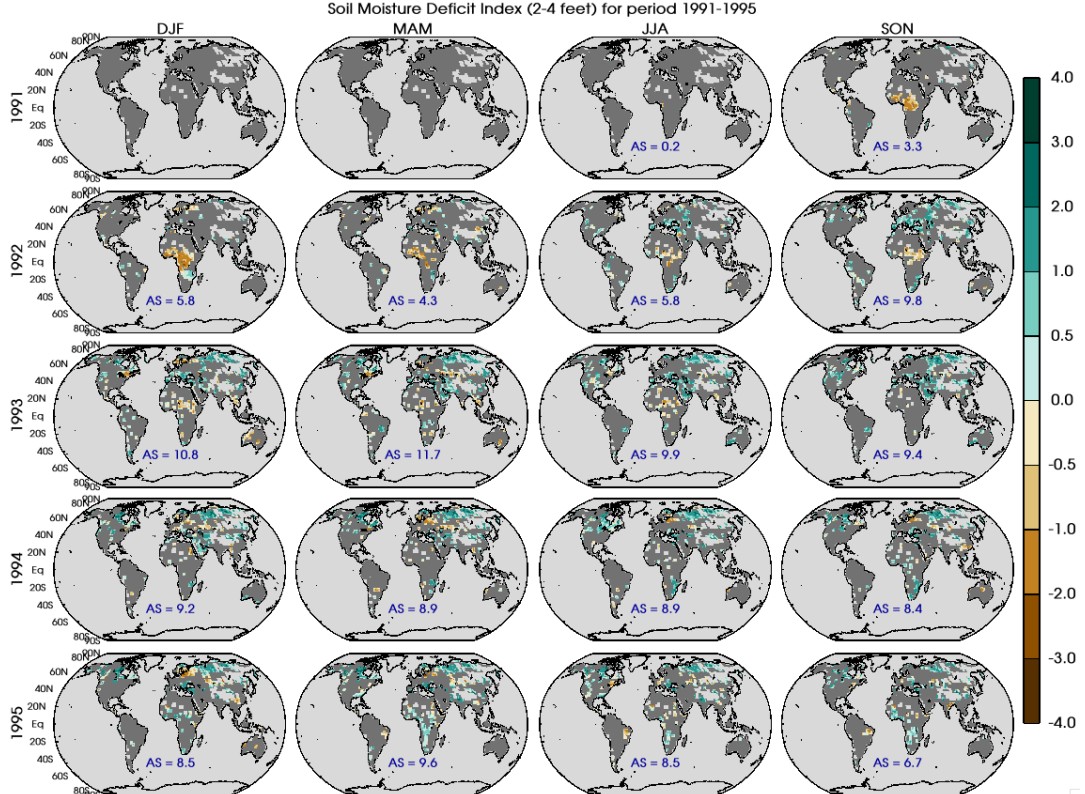

Figure 6. Soil moisture deficit index (SMDI_4) for soil depths between 2-4 feet evaluated
seasonally from 1991 to 1995. Grey color is painted over the grid cell where the SMDI_4 is not
statistically significant in contrast to counter-factual ensemble. The light grey colored regions
represent regions of impermeability. The parameter AS on each panel mark the percentage of
land area which has shown statistically significant dry or wet response after Mt. Pinatubo
eruption.

**3.4.2 Seasonal Evapotranspiration Deficit Index (ETDI)**

As indicated in the methodology section, ETDI calculation is similar to SMDI but is
based on the water stress anomaly which accounts for the difference between actual and potential
evapotranspiration. This is a measure of the flux of water between land and atmosphere and like
SMDI_2 in Figure 6 it shows robust statistical difference over land.



Figure 7 shows that equatorial decreases in ETDI started developing in the DJF season
for the year 1992, and these conditions were persistent over the year. Similar to SMDI_2, ETDI
increases over the region encompassing the Mediterranean and western Asia. However, ETDI
differs from the SMDI_2 over some of the northern hemisphere regions, especially over
Northeastern Asia. A drying response in terms of ETDI in the northern hemisphere regions
persisted during 1993 and 1994, whereas SMDI_2 shows an opposite response. This contrasting
response in terms of ETDI and SMDI_2 points to the complexity of land-atmosphere interactions
over these regions. We utilized model simulated surface energy fluxes (Latent and Sensible heat)
to calculate the potential (PET) and actual evapotranspiration (AET) to estimate the water stress
ratio. In these regions where soil moisture is available in the summer and early winter
months, but a deficit in evapotranspiration reflects the decrease in plant transpiration (latent heat
flux), which may be due to the unavailability of plants. Also, the surface temperature (sensible
heat flux) response supports the non-water-stressed atmospheric conditions and thus overall, it
show a deficit in evapotranspiration. Areas of significant response in terms of ETDI varies from
7 to 14.5% on seasonal basis during the years following the eruption. The largest areas of ETDI
coverage occur during the same time periods as SMDI_2 (between JJA 1992 – JJA 1993).

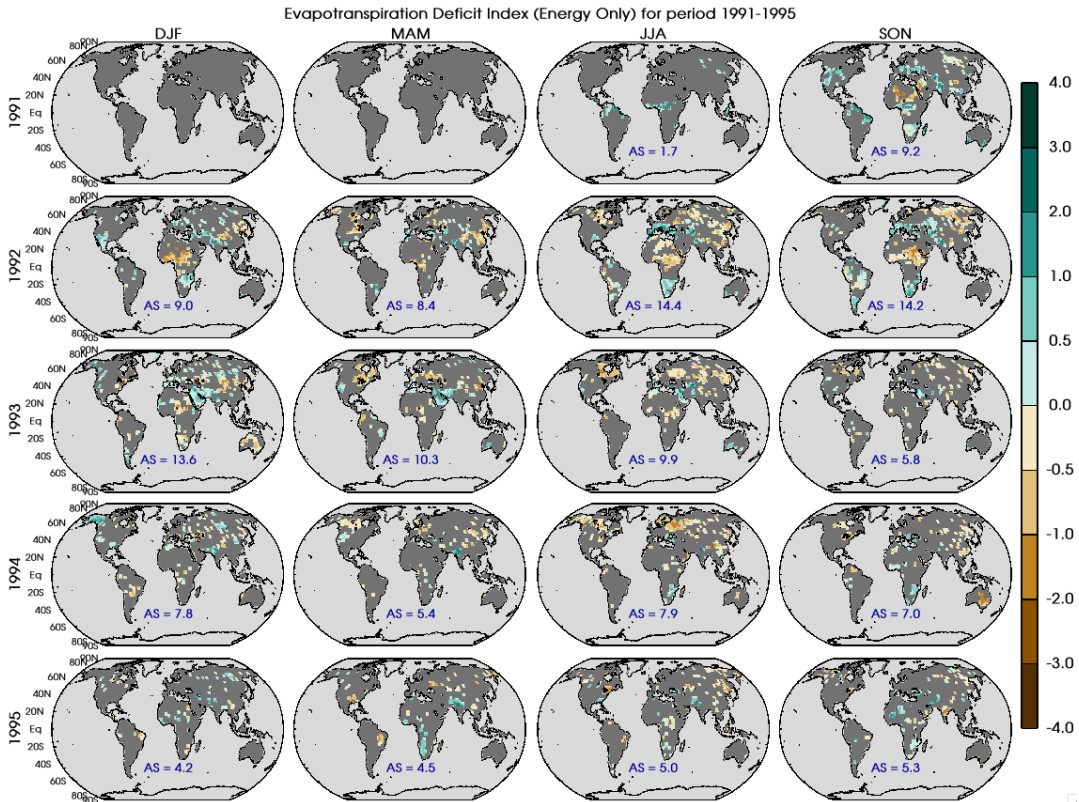

Figure 7. Evapotranspiration deficit index (ETDI) at seasonal scale from 1991 to 1995. Grey color is painted over the grid cell where the ETDI is not statistically significant in contrast to counter-factual ensemble. The parameter AS on each panel mark the percentage of land area which has shown statistically significance dry or wet response after Mt. Pinatubo eruption.

**3.5 Seasonal Plant Productivity Inferences**

SMDI (at depths of 0-2, 2-4, and 4-6 feet) and ETDI have proven helpful in analyzing the climatic impact of the Mt. Pinatubo eruption on a seasonal scale. Additionally, SMDI_2 (top 2 feet) and ETDI have demonstrated elements of a time lag between precipitation on a seasonal scale (Narasimhan and Srinivasan 2005). Crucially, the seasonal depiction of drying/wet conditions via SMDI and ETDI provides a comprehensive overview of prolonged or recurrent





dry/wet conditions in susceptible regions. Moreover, understanding these typical agricultural
drought indices indicates potential effects on plant productivity at the seasonal scale.
Broadly the seasonal responses uncovered an interesting behavior in three more deeply
explored regions. In equatorial Africa, decreases in both SMDI and ETDI indicated that there
was likely a negative impact on plant productivity. On the contrary, the Mediterranean region
(encompassing the eastern Mediterranean and western Asian region) showed increases in SMDI
and ETDI, indicating a positive effect on plant productivity. Northern Asia, on the other hand,
exhibited an increase in SMDI with a decrease in ETDI, indicating that plant productivity likely
decreased, but not because of water-based drivers.
**3.6 High Frequency Impact Pathway Evaluation**
Here we use the model output on a daily scale to calculate weekly drought indices in each
grid cell. These weekly scale drought indices and changes in other atmospheric variables are
explored at the regional scale to understand the associated land-atmosphere interactions in terms
of higher-order impacts. High temporal resolution of these parameters is crucial for analysis of
different stages of the crop cycle in a region. Considering the complexity of the representation of
spatial features, we selected three different regions (shown in Figure 8 and detailed in Table 2) in
the northern hemisphere based on the climate response to Mt. Pinatubo in the seasonal analyses
presented in Section 3.0.  We followed the same strategy described in Section **2.3.1** to mask out
the statistically insignificant grid cells using the counterfactual ensemble after creating the
weekly time series for different drought indices and atmospheric parameters.



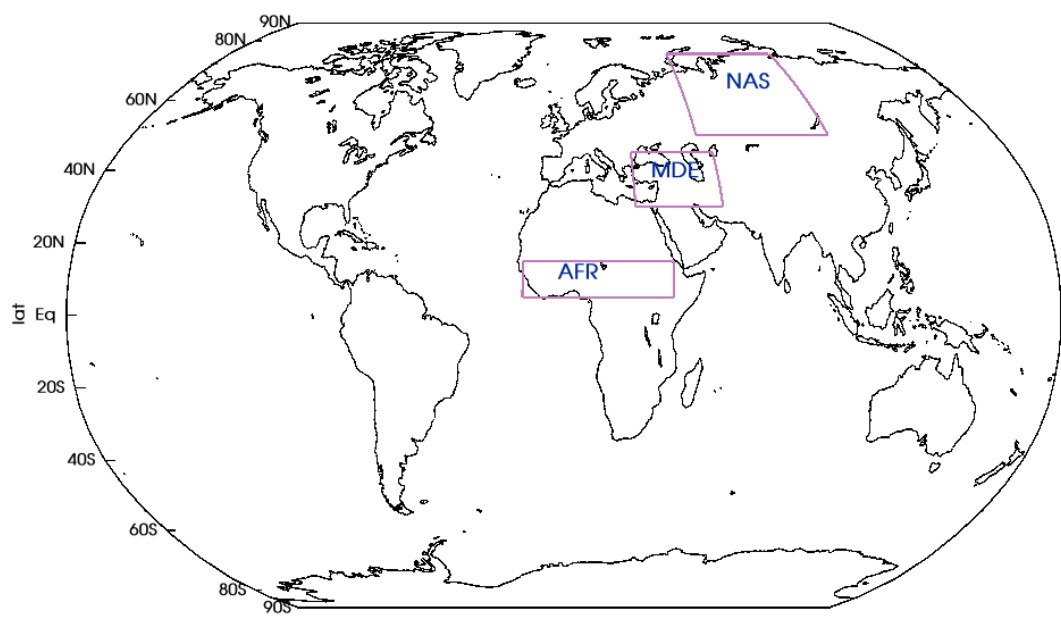

Figure 8: Demarcation of the regions selected over tropics (AFR), mid-latitude (MDE) and high
latitudes (NAS) as shown in table 2.

Table 2. Table showing the details of regions demarcated to regional characteristics at a weekly
scale.

| Sr No. | Region Name | Region Stamp | Lat boundaries | Lon boundaries |
|--------|-------------|--------------|----------------|----------------|
| 1 | Equatorial Africa Region | AFR | 5° N – 15° N | 15° W - 40° E |
| 2 | Middle East Region | MDE | 30° N - 45° N | 27° E - 60° E |
| 3 | Northern Asia Region | NAS | 50° N - 75° N | 55° E - 110° E |




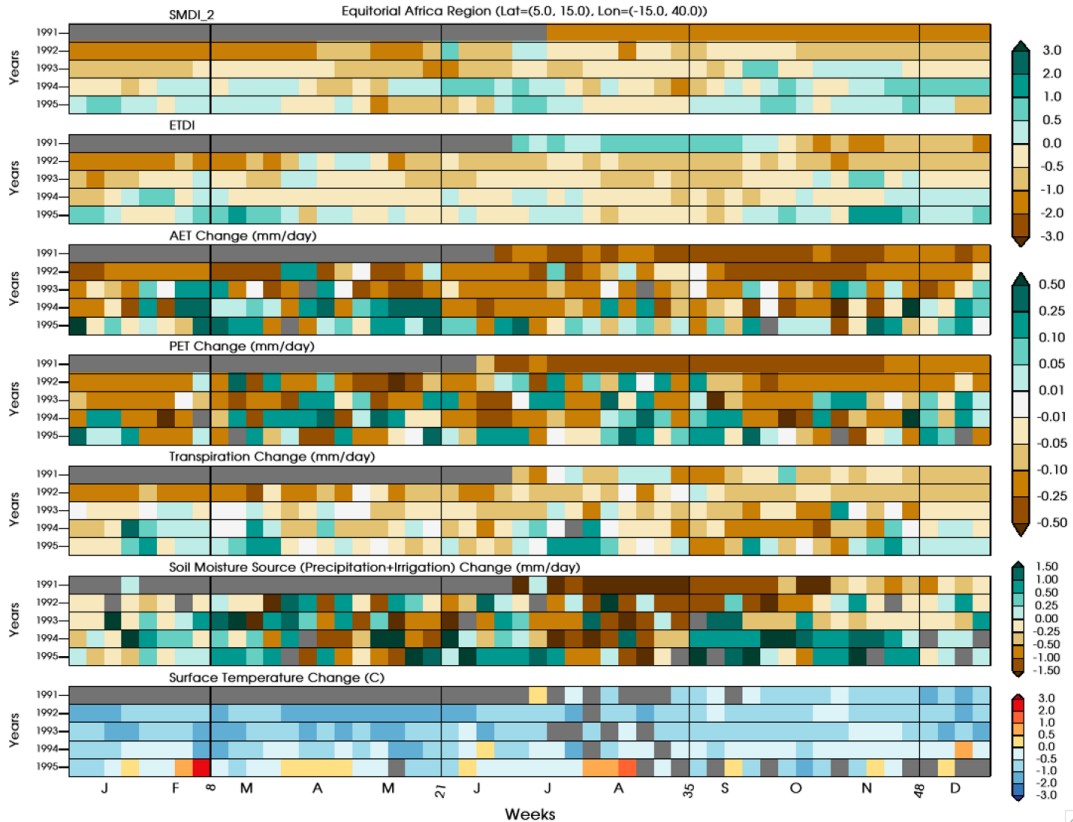

Figure 9. Spatially averaged drought indices (SMDI_2 & ETDI) and anomalies for other drivers (Surface Temperature, Precipitation plus irrigation, Actual and Potential Evapotranspiration, and Transpiration) at weekly scale for the equatorial Africa region (Latitude = 5°-15° N, Longitude = 15° W- 40° E).

**3.6.1 Equatorial Africa**

Figure 9 shows the weekly response to volcanic forcing for the years 1991-1995 in terms of agricultural drought indices (SMDI_2 & ETDI), PET, AET, transpiration, total soil moisture source and surface temperature for an equatorial region in northern Africa. This region exhibits consistent statistical differences across the drivers on a weekly scale and thus the majority of time periods are unmasked revealing the degree of anomaly.





This region lies between the latitude 5-15° N, where the precipitation during the monsoon
season shows a decrease in response to a southern migration of the inter-tropical convergence
zone in energetically deficit northern hemisphere due to volcanic aerosols preferentially reducing
incoming radiation there (Iles et al., 2013; Colose et al., 2016; Singh et al., 2023). Weekly
precipitation change in the equatorial African region shows a significant deficit of more than 1.5
mm per day consistently for several weeks, especially during the JJAS monsoon season. This
region also shows that a deficit in precipitation during the major precipitation season (JJAS) can
result in a soil moisture deficit in the root zone in the following seasons (DJF and MAM in
SMDI_2) and consequently affect the entire crop cycle. The root zone soil moisture, SMDI_2,
also shows a persistent drying through 1993 and combined with the lack of precipitation, the
potential for recharge is limited. Also, this region has no contribution from irrigation as source of
additional soil-moisture as shown in figure S5 (bottom panel) (Cook et al., 2020). Cumulative
annual rainfall change over this region shows a deficit of 33.2, 9.5 and 3.2 mm per day for the
year 1991, 1992 and 1993 and an increase of 10.5 and 13.6 mm per day for the year 1994 and
1995 respectively, where soil-moisture response shows a recovery from the dry conditions.
Hence, it is no surprise that there is a corresponding decrease in ETDI through 1993 that
is consistent with this lack of moisture. However, the evaporative demand, as shown by surface
temperature change, does not consistently decrease until September of 1991 and hence ETDI is
slow to show a decrease in the deficit index. After that point, evaporative demand decreases with
lower temperatures contributing to a decrease in ETDI. However, evapotranspiration is
dominated by transpiration (Seneviratne et al 2010; Nilson and Assmann 2007 and references
therein), and so the majority of the decrease in ETDI is explained by the shown decrease in plant



transpiration. This decrease in plant transpiration is, as expected, well correlated with decreases
in AET.

Conclusively, precipitation response in this region shows dominance in regulating the

ecohydrological conditions. A substantial decrease in the weekly rainfall over the region
perpetuates a root-zone water deficit condition resulting in decreased plant transpiration.
Decreases in both SMDI_2 and ETDI thus indicate the developing agricultural drought
conditions which are confirmed by a decrease in the direct measure of plant transpiration.

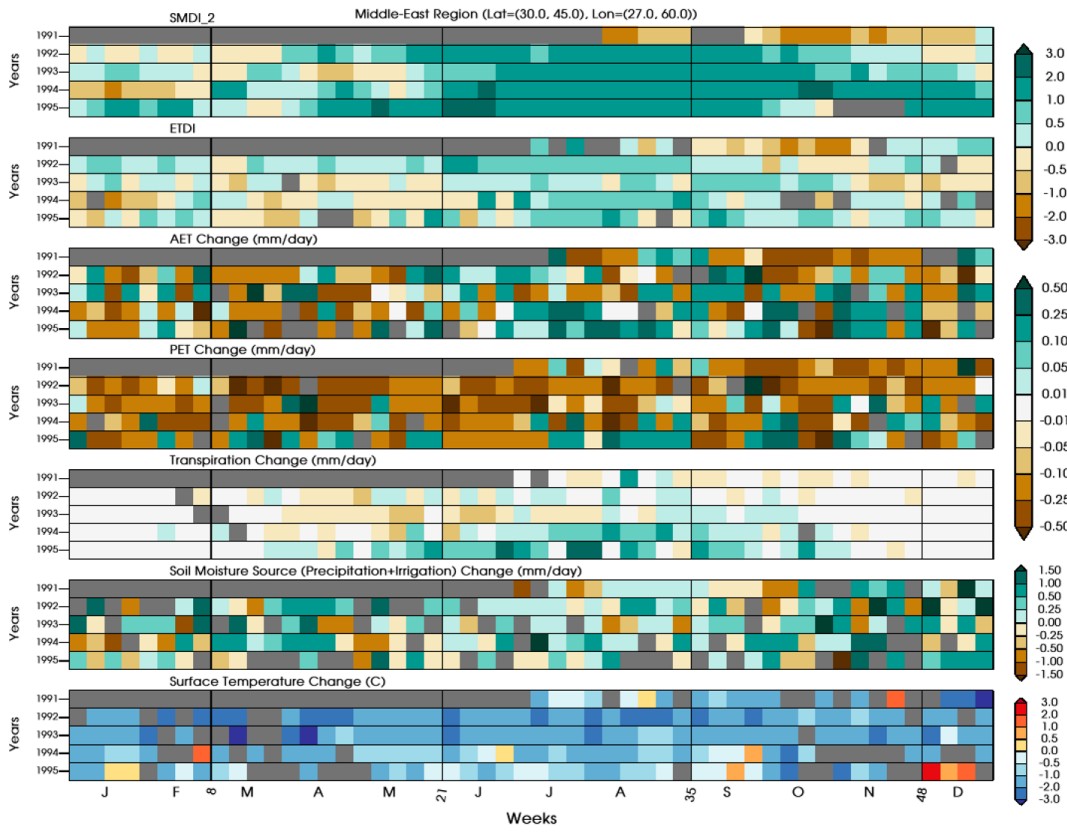


Figure 10. Spatially averaged drought indices (SMDI_2 & ETDI) and anomalies for other drivers
(Surface Temperature, Precipitation plus irrigation, Actual and Potential Evapotranspiration, and
Transpiration) at weekly scale for Middle East (MDE) Region (Latitude =30° N - 45° N,
Longitude=27°-60° E; eastern Mediterranean / western Asian).



**3.6.2 Middle East Region**


Figure 10 shows the region covering the eastern Mediterranean and the western Asian
region where rainfall shows a slight increase after the Mt. Pinatubo eruption. Additionally, this
region exhibits a significantly positive trend in the irrigation practices post-1050, with a
substantial peak over the eastern Mediterranean region following the Mt. Pinatubo eruption
(Cook et al., 2020; Figure 1 and 2).
In the eastern Mediterranean, wet and cold autumns and winters persist for several years
after the Mt. Pinatubo eruption, offering significant root zone recharge potential. The summer
months, in general, reflect a slightly uncertain model response in the regional rainfall with some
weeks of deficit and some of excess but an additional supply of water through the irrigation
contributes to the overall moisture content in the region (Figure S5). Root zone soil moisture,
SMDI_2, shows ample water during the growing seasons through the entire analysis period.
Taken together, it is clear that this region is not moisture-limited and there is sufficient
precipitation and irrigation supply to recharge root zone moisture as plants grow. Cumulative
weekly anomalies show that precipitation change in 1991 is slightly negative (-0.5 mm per day)
but an increase in annual rainfall of 13.8, 8.0, 10.9, and 4.5 mm per day is simulated for the year
1992, 1993, 1994 and 1995 respectively. Implemented irrigation over this middle east region
shows a strong positive trend for the period 1950-2005 (Cook et al., 2020), and a substantial
cumulative increase of 0.5, 1.3, 1.3, 0.8 and 0.9 mm per day in the irrigation for the years 1991
to 1995 serves as the additional source of moisture supply over the region (Cook et al., 2020).
Thus, irrigation supplies water especially in summer months when rainfall change shows a few
weeks of deficit and contributes 10-20% of the soil-moisture source change (Figure S5).



The corresponding increases in ETDI and AET reflect the ample source of water
available for transpiration in the region. Transpiration is again temporally correlated with AET,
but the increases are less well pronounced. At the same time, there is a decrease in PET response
correlated with the stronger decrease in temperature in this region as compared to equatorial
Africa. The decrease in PET coupled with the increase/maintenance in AET (through
transpiration) combine to result in increased ETDI. Thus, in general, agricultural productivity is
positively affected in this region as there is ample moisture to support it. However, there are still
heterogenous patterns in this data showing that 1993, for instance, may have had some impact on
plant productivity with positive but lower magnitude ETDI, inconsistent AET, and decreased
transpiration.
Regardless of the presence of volcanically induced response or not, the weekly scale
analysis demonstrates its importance by virtue of an example from the year 1993, where rainfall
deficit is produced during the 15th, and 16th weeks (April) of the year. Combined with low
SMDI_2, this could result in a lack of moisture availability during a crucial stage of crop cycle.
Given the duration, this could significantly influence the overall seasonal crop production.  Thus,
even if the majority of the crop cycle possess favorable conditions, negative impacts at essential
phases of the crop cycle can crucially affect production in ways that seasonal averages would be
unable to reveal.



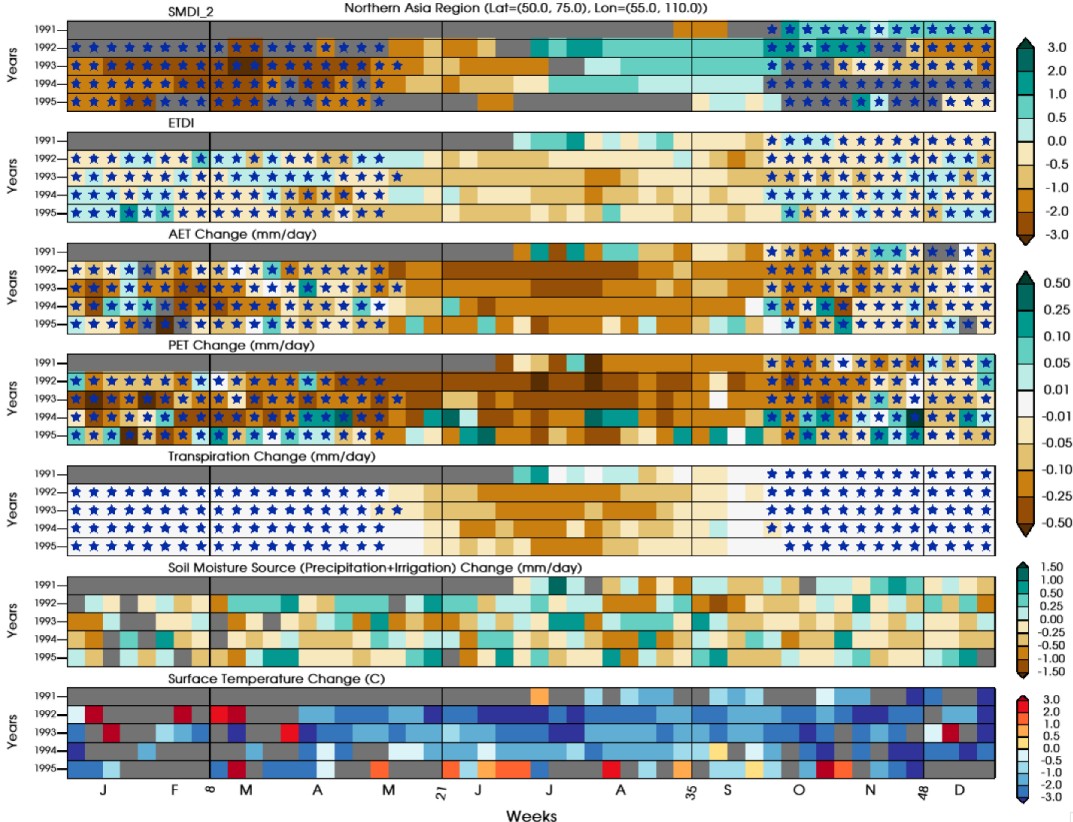


Figure 11. Spatially averaged drought indices (SMDI_2 & ETDI) and anomalies for other drivers
(Surface Temperature, Precipitation plus irrigation, Actual and Potential Evapotranspiration, and
Transpiration) at weekly scale Northern Asia Region (Latitude =50° N - 75° N, Longitude=55°-
110° E).  Blue stars represent the weeks with average surface temperature below freezing point.

### 3.6.3 Northern Asia

Finally, we selected a region (NAS) in higher latitudes to explore the interplay between

the various drivers governing the conditions for plant productivity. Again, this region exhibits

consistent statistical differences across the drivers on a weekly scale.  However, with higher

latitudes also comes strong seasonal controls over plant productivity with below freezing

temperatures, shown with blue stars, halting productivity. Hence our analysis here will focus on

months during which plants can grow (~MJJAS).



665   Precipitation changes are highly uncertain over the entire analysis period, but in general,

666 there is a slight trend towards increased precipitation from Nov 1991- June 1992 followed by

667 decreased precipitation through 1994.  As shown in Fig S5, the irrigation contribution to soil

668 moisture in this region is negligible. Cumulative weekly precipitation anomalies show an annual

669 increase of 0.05 mm per day for year 1991 and decrease of -2.0, -4.1, -6.8 and -1.0 mm per day

670 for the years 1992 to 1995 respectively. Alternatively, root zone moisture shows ample water

671 available to plants during the JAS growing months after a strong deficit in the early MJ months.

672 Certainly, in these summer months the melting of frozen surfaces and snow supplies moisture in

673 the upper layers to become wet accounting for this strong dichotomy.

674   However, there are not corresponding increases in ETDI and AET after the 1991 season.

675 This indicates that even though there is ample water, plants are still not growing; this is

676 conclusively confirmed by the decrease in transpiration starting in 1992. Meanwhile, the

677 simultaneous decrease in PET response is correlated with the strongest decrease in temperature,

678 on the order of 2-3° C, for the three regions.

679   Unlike the other two regions for which SMDI_2 and ETDI exhibited similar wet/dry

680 patterns, this location shows diverging patterns. Broadly, this reveals that even though there is

681 moisture to support plant productivity, the moisture is not being utilized. Hence, other factors

682 must be the cause of decreased plant transpiration and ETDI. The stronger decrease of PET

683 compared to AET indicates that temperature may be playing a role here. Temperature is a direct

684 proxy of decreased incident radiation. Hence, combined temperature and radiation effects are

685 likely the most important controls on decreased plant productivity in this region, not moisture

686 conditions.






**4.0 Conclusions**

This study has used the Earth system modelling framework to explore the mechanisms by

which the 1991 Mt. Pinatubo eruption affected the hydroclimatic conditions and water-based
drivers of plant productivity. NASA GISS ModelE2.1 with the interactive chemistry and aerosol
microphysical module (MATRIX) has demonstrated a successful simulation of microphysical
properties (effective radius of order ~0.5 μm, aerosol extinction of ~0.21) of volcanic aerosol
with induced radiative effect of longwave, shortwave, and net forcing of order of +3 Wm$^{-2}$, 8
Wm$^{-2}$ and -5 Wm$^{-2}$ respectively.  This is consistent with the observations and other estimates
(Russell et al., 1996; Bingen et al., 2004; Stenchikov et al., 1998; Bauman et al., 2003, Lacis et
al., 1992, Lacis 2015; Stenchikov et al., 1998; Hansen et al., 1992; Minnis et al., 1993; Brown et
al 2024). The temperature response pathway shows Mt. Pinatubo eruption affected global surface
cooling by ~0.5 °C with corresponding tropical lower stratosphere warming of 2-3 °C for several
years after the eruption.  This is consistent with the observations and other modeling estimates
(Hansen et al., 1996; Parker et al., 1996; Stenchikov et al., 1998; Minnis et al., 1993; Kirchner et
al., 1999; Ramachandran et al., 2000; Dutton and Christy 1992; Brown et al 2024). The GISS
model simulates regional patches of decreases in rainfall of the order of 1 mm per day over the
tropics and northern hemisphere regions (consistent with Joseph and Zheng, 2011; Liu et al.,
2016; Trenberth and Dai, 2007), but the overall response of rainfall is highly uncertain.  This
study has endeavored to explore the secondary impacts of a volcanic eruption beyond the
changes in radiation and temperature by examining agricultural drought indices to better infer
impacts to plant productivity. Droughts are among the prime factors affecting regional crop yield
at any stage of the crop cycle (Ben Abdelmalek and Nouiri, 2020; Leng and Hall, 2019; Raman
et al., 2012). Both SMDI and ETDI represent the developing short- and longer-term conditions



which support plant productivity, especially for agricultural applications. SMDI represents
excess/deficit of soil moisture in different layers, whereas ETDI represents the active interaction
between the land and atmosphere under perturbed climate conditions. An increase in the gap
between the potential evapotranspiration (PET) and actual evapotranspiration (AET) represents
the increased water stress condition either by increased potential evapotranspiration (water
demand) or by the decrease in water available for evapotranspiration (lower AET). These
drought indices confirm the moisture source based dry and wet pattern in early 1992 and the
following years over the tropical and northern hemispheres mid-latitude regions correspondingly
as a response to the volcanic forcings due to the Mt. Pinatubo eruption. Using both drought
indices, we conclude that approximately 10-15% of land region shows statistically significant
dry or wet patterns in the volcanically perturbed climate conditions for 1992 and 1993. The
fraction of land region showing a significant dry or wet response range between 5-10% for the
next two (1994 and 1995) years.  Broadly, the seasonal responses uncovered interesting behavior
in three regions which we explore more deeply.  In equatorial Africa, decreases in both SMDI
and ETDI indicated that there was likely a negative impact on plant productivity while in a
contrary manner the Middle East region showed increases in SMDI and ETDI indicating a
positive impact on plant productivity.  Northern Asia in comparison exhibited an increase in
SMDI with a decrease in ETDI indicating that plant productivity likely decreased, but not
because of water-based drivers.

Using these key pattern differences as motivation, we deepened our analysis of these

drought indices using higher temporal (weekly) frequencies and by incorporating AET, PET, and
transpiration directly.  In general, these regional analyses possess much stronger statistical
significance on the weekly scale, and they further confirmed the seasonally based inferences



above. Further, weekly drought indices show the temporal variability characteristics in the
signal, which also demonstrates the utility of explaining the effectiveness of short-term dry/wet
conditions corresponding to a regional crop cycle. In locations where there is insufficient/excess
soil moisture, there is a corresponding decrease/increase in evapotranspiration (AFR/MDE) and
hence decreased/increased plant productivity.

This work is the first to conclusively show that there is an excess of root-zone soil

moisture in high latitudes (NAS) which is not being utilized by plants to grow establishing the
main control is likely temperature and radiation based confirming the results of (Krakauer and
Randerson, 2003) and (Dong and Dai, 2017). The intricate nature of the compounded response,
particularly regarding the soil moisture-based impact pathways in tropical regions and higher
latitudes across the northern hemisphere, also underscores the necessity of broadening the scope
of the investigation beyond soil moisture and land-atmosphere interactions. The current setup of
the NASA GISS model effectively runs using prescribed vegetation with static plant functional
types and leaf area index, and the inclusion of dynamic vegetation could be crucial for adding
interactive land surface responses. Also, assessing the influence of the regional and local biome
on photosynthesis rate could provide a more detailed understanding of how these processes
specifically respond to the climate impact of volcanic forcings. McDermid et al. (2022) have
demonstrated the sensitivity of regional hydroclimate to the local changes in soil organic carbon
changes using the soil moisture content. The results presented in this study in terms of soil-
moisture-based drivers to the plant productivity and surface temperature response in the northern
hemisphere high latitudes also hint towards the dominance of temperature effects on enhanced
carbon sink in terms of soil and plant respiration and reduced NPP (Krakauer and Randerson,
2003; Lucht et al., 2002). Meanwhile, water-based drivers dominate productivity responses in



multiple tropical and sub-tropical regions. Our results illustrate that soil-moisture-based
conditions in the different regions can be useful for evaluating and understanding the full impacts
on the agricultural yield and regional carbon sink response if the dynamic vegetation and crop
cover changes. A recently developed fully demographic dynamic vegetation model (ModelE-
BiomE v.1.0 (Weng et al., 2022)) with interactive biophysical and biogeochemical feedback
between climate and land systems for NASA GISS ModelE could be helpful in evaluating the
carbon cycle response under such forcings.
**Code/Data availability.**
Details to support the results in the manuscript is available as supplementary information is
provided with the manuscript. GISS Model code snapshots are available at
https://simplex.giss.nasa.gov/snapshots/ (National Aeronautics and Space Administration, 2024)
and calculated diagnostics are available at zenodo repository
(https://zenodo.org/records/12734905) (Singh et al., 2024). However, raw model output and data
at high temporal (daily) resolution and codes are available on request from author due to large data
volume.
**Acknowledgements**
Resources supporting this work were provided by the NASA High-End Computing (HEC)
Program through the NASA Center for Climate Simulation (NCCS) at Goddard Space Flight
Center. The authors thank Ben I Cook, Nancy Y Kiang, Igor Aleinov and Michael Puma for their
input through multiple discussions with the project members. RS, KT, DB, LS, BW, and KM were
supported by the Laboratory Directed Research and Development program at Sandia National
Laboratories, a multi-mission laboratory managed and operated by National Technology and
Engineering Solutions of Sandia LLC, a wholly owned subsidiary of Honeywell International Inc.
for the U.S. Department of Energy's National Nuclear Security Administration under contract DE-
NA0003525. This paper describes objective technical results and analysis. Any subjective views
or opinions that might be expressed in the paper do not necessarily represent the views of the U.S.
Department of Energy or the United States Government.
**Author's contributions**





RS, KT, DB, LS and KM identified the study period in consultation with the other authors and RS, KT, DB, LS and BW designed the underlying simulations strategies. RS and KT implemented it and performed the simulations using NASA GISS ModelE. RS and KT have performed the analysis. RS created the figures in close collaboration with all authors. RS wrote the first draft of the manuscript, and all other authors has contributed the writing of subsequent drafts. All authors contributed to the interpretation of results.

**Competing interests**

One of the co-authors is member of the editorial board of Atmospheric Chemistry and Physics.

**Short Summary**

Analysis of post-eruption climate conditions using the impact metrics is crucial for understanding the hydroclimatic responses. We used NASA's Earth system model to perform the experiments and utilize the moisture-based impact metrics and hydrological variables to investigate the effect of volcanically induced conditions that govern plant productivity. This study demonstrates the Mt. Pinatubo's impact on drivers of plant productivity and regional and seasonal dependence of different drivers.

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
