# Peer review of "Mount Pinatubo's effect on the moisture-based drivers of plant productivity"

_EGUsphere, 2024_

## Referee Comment (RC2)

**Review of "Mount Pinatubo's effect on the moisture-based drivers of plant productivity „by Ram Singh et al.**

The aim of the current study is to understand how volcanic eruptions affect ecohydrological conditions and plant productivity. The authors used the NASA Earth System Model to simulate the 1991 eruption of Mt. Pinatubo and to detect its response on soil moisture and evapotranspiration as short-term ecohydrological controls on plant productivity. Using the Soil Moisture Deficit Index (SMDI) and the Evapotranspiration Deficit Index (ETDI), they find that about 10-15% of the land area shows statistically significant dry or wet patterns in the volcanically perturbed climate conditions for 1992 and 1993, and between 5-10% in the following years (1994-1995). The authors focus on three regions that show different responses in EDTI and SMDI. In Equatorial Africa, decreases in both indicate a likely negative impact on crop productivity, while in the Middle East region increases indicate a positive impact on crop productivity. North Asia on the other hand, shows an increase in SMDI and a decrease in ETDI, indicating that crop productivity has probably decreased, but not due to water-related factors.

The paper needs some major improvements. It needs to be more streamlined and the results should be discussed in a broader context, see general comments below. I therefore recommend publication only after major revisions.

General comments

- Part of the paper reads like a model evaluation paper of the climate response to volcanic forcing in the MATRIX version of the GISS ModelE2.1 (Bauer et al., 2020). I am therefore confused as to the purpose of this paper. If validation of the Pinatubo simulation is one of the aims of the paper, this should be clearly communicated. The evaluation of the primary dependent variables temperature and precipitation is quite lengthy and has been done before in other contexts, see points below.
    - Regarding the evaluation, I wonder why you do not put your aerosol microphysical model results into a broader context and relate them to recent work. Quaglia et al (2023) published last year an extensive multi-model data comparison of the Pinatubo episode with different aerosol microphysical models.
    - The GISS E2.1 model has been used before to study the impact of volcanoes on climate, so this aspect is not really new. I wonder why you do not discuss the climate response after the Pinatubo eruption in your model version with that simulated with the CMIP6 version of the GISS ModelE2.1 (Kelly et al. 2020) in the historical runs and in the Pinatubo VolMIP ensemble (Weierbach et al., 2020). From my point of view, the only open/interesting question here is I how does the surface climate response change when you calculate the aerosol microphysics online.
    - o Related to this. I do not understand why you need an aerosol model for your study of the impact of volcanic forcing on moisture-based drivers of plant productivity. This study would also work with prescribed volcanic forcing: you could use the historical CMIP6 ensemble (Miller et al., 2021) or the 81-member VolMIP Pinatubo ensemble (Weierbach et al., 2023). For these simulations, you would have more than 11 ensembles, which would make your results even more statistically robust.
- The authors are right that there are not many studies on this recent topic. However, soil moisture changes due to volcanic eruptions have been discussed in the context of volcanic impacts on the carbon cycle (e.g. Fröhlicher et al., 2011). There are also some interesting discussions in Zuo et al. (2019a ,b) on the hydroclimate response after a volcanic eruption, where not soil moisture but other relevant hydroclimate parameters are related to NPP. Furthermore, there is a broad discussion in the geoengineering community about the impact of stratospheric aerosol on soil

moisture and food production for solar geoengineering, see for example Cheng et al. (201). These issues should be addressed in the paper.

- The authors speculate a lot in the paper about potential impacts on crop productivity, but they have not shown any GPP or NPP anomaly plots. I wonder why, as this would strengthen the paper considerably.

Specific comments

| | |
|---|---|
| Line 23-24: | Could be deleted. if model evaluation is not a specific subject of your paper. |
| Line 27: | You do not show any agricultural response in the paper so please be careful with your wording. |
| Line 28: | Not clear what you mean with "these higher-order impacts". |
| Lines 41-44 | Too many references for something well known and obvious. I suggest that you refer here instead to some overview papers e.g. Marshall et al (2022), Kremser et al (2016), Timmreck et al (2012). |
| Line 48: | Again, you can reduce the amount of citations and refer to some overview papers or the recent model intercomparison paper. |
| Line 53 ff: | See point above. |
| Lines 73 ff: | Please cite here in addition or instead of Toohey et al (2016) the most recent paper for the mid-6th century volcanic impact on post-volcanic climatic and societal response over Scandinavia: van Dijk et al (2023). |
| Lines 193-195: | It is not clear here why you use the climatology of the years 1950-2014, maybe you refer here already to the supplementary material. |
| Line, 227: | Index instead of Indices |
| Lines 239-240: | Please use SI units, even its not typical for SDI, you can put the "feet metric" in brackets |
| Line 247ff: | Here and in the following lines the reference style seems not correct. |
| Lines 284-286: | Concerning the justification, it is quite difficult to see in the supplementary Fig S1 the difference between the three reference periods. Maybe a difference plots between them would more useful. |
| Lines 309-311: | This sentence sounds strange, please reformulate. |
| Lines 357-370: | It is not clear to me why the lower stratospheric temperature response is relevant for your specific topic |
| Lines 416 -421: | So why choose for your analysis few realizations with prognostic aerosol instead of more realizations with many members? |
| Line 436: | Section 3.4. |
| Lines 444-446: | This sentence is not clear, please reformulate. |
| Line 535: | What are "elements of a time lag between precipitation"? |
| Lines 644 ff: | Does this happen in all realizations or is it just a coincidence in the ensemble mean? How meaningful are changes of individual weeks? |
| Lines, 691 ff: | I think you can shorten the part about the model performance/evaluation substantially. You need not to list the references here when you already have included them in the text |
| Line 719: | "volcanic forcings due to the Mt. Pinatubo eruption" sounds strange, please revise |

- Tables:
  - Table 1 can be merged with the figure caption of Figure 8

- Figures:

- In general: The multi-panel figures (5, 6,7) are too small and hard to read and therefore not very convincing. I strongly recommend to reduce the number of panels by either showing only specific years or specific seasons. This would make the figures much more readable and therefore much better emphasize the point. The missing panels (seasons, years) could be put into the supplementary material.
- Figures 3, 4 : 1st two panels are useless and could be deleted.

- References:
  - Please update the reference to Brown, H. Y, Geosci. Model Dev., 17, 5087–5121, https://doi.org/10.5194/gmd-17-5087-2024, 2024.

**References**

Bauer, S. E., Tsigaridis, K., Faluvegi, G., Kelley, M., Lo, K. K., Miller, R. L., Nazarenko, L., Schmidt,G. A., and Wu, J.: Historical (1850–2014) Aerosol EvoluDon and Role on Climate Forcing Using the GISS ModelE2.1 ContribuDon to CMIP6, Journal of Advances in Modeling Earth Systems, 12, e2019MS001978, h\ps://doi.org/10.1029/2019MS001978, 2020.

Cheng, W et al. Soil moisture and other hydrological changes in a stratospheric aerosol geoengineering large ensemble. Journal of Geophysical Research: Atmospheres, 124, 12,773–12,793 https://doi.org/10.1029/2018JD030237, 2019.

Frölicher, T. L., Joos, F., and Raible, C. C.: Sensitivity of atmospheric $CO_2$ and climate to explosive volcanic eruptions, Biogeosciences, 8, 2317–2339, https://doi.org/10.5194/bg-8-2317-2011, 2011.

Kelley, M. et al.: GISS-E2.1: ConfiguraDons and Climatology, Journal of Advances in Modeling Earth Systems, 12, e2019MS002025, https://doi.org/10.1029/2019MS002025, 2020.

Kremser, S., et al.: Stratospheric aerosol – Observations, processes, and impact on climate, Rev. Geophys., 54, 1–58, https://doi.org/10.1002/2015RG000511, 2016.

Marshall, L, E C. Maters, A. Schmidt, C. Timmreck, A. Robock, and M. Toohey, Volcanic effects on climate: recent advances and future avenues. Bull Volcanol 84, 54. https://doi.org/10.1007/s00445-022-01559-3, 2022.

Miller, R.et al.: CMIP6 historical simulations (1850–2014) withGISS-E2.1, J. Adv. Model. Earth Syst., 13, e2019MS002034, https://doi.org/10.1029/2019MS002034, 2021

Quaglia, I., Timmreck, C., Niemeier, U., Visioni, D., Pitari, G., Brodowsky, C., Brühl, C., Dhomse, S. S., Franke, H., Laakso, A., Mann, G. W., Rozanov, E., and Sukhodolov, T.: Interactive stratospheric aerosol models' response to different amounts and altitudes of SO2 injection during the 1991 Pinatubo eruption, Atmos. Chem. Phys., 23, 921–948, https://doi.org/10.5194/acp-23-921-2023, 2023.

Timmreck C. (2012), Modeling the climatic effects of volcanic eruptions, Wiley Interdisciplinary Reviews: Climate Change doi: 10.1002/wcc.192.

Toohey, M., Krüger, K., Sigl, M., Stordal, F., and Svensen, H.: ClimaDc and societal impacts of a volcanic double event at the dawn of the Middle Ages, ClimaDc Change, 136, 401–412, https://doi.org/10.1007/s10584-016-1648-7, 2016.

van Dijk, E., I. Mørkestøl Gundersen, A.de Bode, H. Høeg, K. Loftsgarden, F. Iversen, C. Timmreck, J. Jungclaus and K. Krüger (2023). Climatic and societal impacts in Scandinavia following the 536 and 540 CE volcanic double event, Clim. Past, 19, 357–398, https://doi.org/10.5194/cp-19-357-2023, 2023.

Weierbach, H., LeGrande, A. N., and Tsigaridis, K.: The impact of ENSO and NAO initial conditions and anomalies on the modeled response to Pinatubo-sized volcanic forcing, Atmos. Chem. Phys., 23, 15491–15505, https://doi.org/10.5194/acp-23-15491-2023, 2023.

Zanchettin, D et al.: Effects of forcing differences and initial conditions on inter-model agreement in the VolMIP volc-pinatubo-full experiment, Geosci. Model Dev., 15, 2265–2292, https://doi.org/10.5194/gmd-15-2265-2022, 2022.

Zuo, M., Zhou, T., and Man, W.: Hydroclimate Responses over Global Monsoon Regions Following Volcanic Eruptions at Different Latitudes, J. Climate, 32, 4367–4385, https://doi.org/10.1175/jcli-d-18-0707.1, 2019a

Zuo, M., Zhou, T., and Man, W.: Wetter Global Arid Regions Driven by Volcanic Eruptions, J. Geophys. Res.-Atmos., 124, 13648–13662, https://doi.org/10.1029/2019jd031171, 2019b.

---

## Author Comment (AC1)

The submitted manuscript examines how the 1991 Mt. Pinatubo eruption impacted hydroclimatic conditions and water-related drivers of plant productivity through a series of Earth system model simulations. Unlike previous studies that mainly focused on radiation and temperature changes, according the authors, this work emphasizes the secondary impacts of volcanic eruptions by analyzing agricultural drought indices (SMDI and ETDI) to infer their effects on plant productivity. These indices reveal distinct moisture-driven dry and wet patterns in early 1992 and subsequent years over tropical and mid-latitude regions of the Northern Hemisphere, linked to volcanic forcing from the eruption. The authors focus on three regions and argue that insufficient/excess soil moisture leads to corresponding decreases/increases in evapotranspiration and plant productivity. In high-latitude areas with excessive root-zone soil moisture, temperature and radiation appear to play a key role in determining plant growth.

Overall, while the manuscript contains some but very limited novel insights, it is poorly written, riddled with typos, and difficult to follow. The lack of attention to formatting suggests the manuscript was submitted in an early draft state. I strongly recommend that *Atmospheric Chemistry and Physics* consider this manuscript only after at least significant revisions.

Thank you for your valuable comments, which are very helpful in improving the overall presentation of the manuscript. We acknowledge that certain paragraphs of the manuscript were poorly written and have rephrased them (including, but not limited to, sentences in the sections listed below). Additionally, the manuscript has been substantially revised in response to comments and suggestions from Reviewer 2 and co-authors. Beyond addressing specific comments, a brief summary of the changes made in the revised manuscript is provided below. Responses to specific comments are in blue text under each comment, with italicized text indicating specific additions to the manuscript. Pointers to specific changes in the main manuscript are highlighted in red brackets.

Summary of additional changes made in manuscript (Please refer to response to Reviewer 2).

1.) Rephrasing of sentences has been done across the abstract, Section 1 (Introduction), Section 2.2 (Experimental Design), Section 2.3 (Method), Section 3.2 (Aerosol Dispersion), the entire Section 3 (Results), and Section 4 (Conclusion). Please refer to the track-changes version of the manuscript.
2.) The introduction section has been revised to incorporate relevant findings and rephrased extensively. Similarly, other sections have also been updated.
3.) The experimental design section has been revised to include a detailed description of the experiment flow, and Table 1 has also been updated.
4.) The method section has been enhanced by simplifying the equations and providing additional clarifications.
5.) The evaluation of the microphysical properties of volcanic aerosols has been relocated to the supplementary section (S2.0), along with Plot 1 (radiative forcings).
6.) Plot 2 from the submitted version has been modified by removing the panel for lower stratosphere temperature (MSU-TLS), and the corresponding description has been moved to the supplementary section..
7.) Plots 4, 5, 6, and supplementary figure S6 have been revised to magnify the land surface in their panels..
8.) Table 2 has been removed, and the details of the regions have been incorporated into Figure 7.
9.) Supplementary Plot S2 has been added to provide context for the selection of the baseline period and to describe the alternative approach..

10.) A paragraph discussing the GPP response has been added to the conclusion section, and Plot S9 has been included in the supplementary information. First paragraph of conclusion section is also modified.

Major concern:

Lines 347-349: Why did the authors choose to show anomalies of PCH relative to the climatology from 1950-2014 rather than comparing simulations with and without the Mt. Pinatubo eruption?

Since we use the climatology as the baseline for many of the metrics (i.e., calibration period for SMDI, ETDI) presented in the manuscript, we chose to remain consistent for anomalies for PCH, too. Additionally, the mean climate of Pinatubo (PCH) and no-Pinatubo (NP) are used to perform the paired t-test to demarcate the regions of significantly (95% confidence level) different climates. However, the suggested approach by the reviewer can be a viable alternative. We now show this in the manuscript with an example of surface temperature change in the supplementary information (Figure S2, along with the section S1.0). This shows that the spatial pattern of the surface temperature response to the Pinatubo eruption is qualitatively the same with the approach used in the manuscript, where a baseline period is used. Regional differences are larger at some places with this new approach, but overall the results across the two methods are very similar.

We modified the relevant paragraph as follows (line 207-217):

*"However, directly comparing the difference between the two ensembles (PCH and NP) is an alternative approach to presenting the Pinatubo effect (see Supplement Figure S2). Using either approach leads to the same general conclusions, with only small quantitative differences. Nevertheless, we chose to remain consistent with the baseline requirements for other metrics as well and used the historical climatology for period 1950-2014 as the baseline for the core of our analysis. The coloring emphasizes the significant regions of anomalies. But we also emphasize the difference in calculations: the grey areas show no significant change between the PCH and NP ensembles, while the anomalies are PCH ensemble mean minus climatology."*

Specific Comments:

Line 25: Please clarify what "dominating variability" refers to.

We re-phrased the sentence (line 25):

*"The rainfall response is spatially heterogenous with large temporal variability, yet still shows suppressed rainfall in the northern hemisphere after the eruption".*

Lines 26-27: Is it typical to group both wetting and drying (increase and decrease) under "statistically significant agricultural response"?

This study is more focused on presenting the change in climate state which affects the plant productivity irrespective to the direction of change. This is why we grouped the opposite responses of drying and wetting together, to state the percentage of area which shows a

statistically significant change in terms of the indices utilized to detect the response of Pinatubo eruption. The details about the direction of change (wet or dry) are discussed in detail in the results section (lines 26-27 of the submitted manuscript).

We modified the sentence to emphasize that both signs are considered in this metric:

Line 26-28

*"We find that up to 10-15% of land regions show a statistically significant hydroclimate response (wet and dry) as calculated by the Soil Moisture Deficit Index (SMDI) and Evapotranspiration Deficit Index (ETDI)"*

Lines 67-68: The phrase "NPP (Net Primary Productivity)" should be revised to "Net Primary Productivity (NPP)."

Modified as suggested (line 64-65)

Line 95 and elsewhere: Citing literature using "and references therein" is uncommon. Consider revising.

We modified this citation formatting and included the proper citations.

Line 141: Please provide the full name of MODIS.

Spelled-out the acronym (line 145).

Lines 153, 247, 254, 257, etc.: Citation formats are uncommon. Please standardize them.

Fixed the use of parentheses.

Lines 170-173: Verify whether it is accurate to state that 15 Tg of the total ~15.2 Tg $SO_2$ was emitted on June 15th.

Daily emission of SO2 is supplied as input to the model as provided by (Carn et al., 2016). The inventory gives an emission of 140.7, 54.06, 15000.7 and 0.070 kt $SO_2$/day for the 13th, 14th , 15th and 16th of June, respectively, which means that our numbers are correct.  We fixed the citation, which was incorrect.

Line 176-178

*"The cumulative Mt. Pinatubo emission is 15194 kt (~15.2 Tg) of $SO_2$ injected from 13th to 16th of June 1991 above the Mt. Pinatubo vent, with a maximum of 15000 kt (15 Tg) emitted on June 15th at a plume height of 25 km Carn et al., (2016)."*

Line 176 or Table 1: Describe the experiments in the text instead of solely relying on the table.

We described the experimented protocol in text along with the relevant details in the modified Table 1.

The following is the text added to the paragraph describing the experiment design.

(line 181-194)

"*The model simulations we performed (Table 1) are described here. We started from the 1400-year-long preindustrial control run from CMIP6 (GISS-CMIP6-PI) with the prescribed average AOD historical period, which is further extended for 500 years using the GISS ModelE2.1 – MATRIX with prognostic tracers (GISS-PI). Then, the CMIP6 historical run (GISS-HIST-SO2; 1850-2014) started with all forcings as specified by CMIP6 except the daily emission rate of injection of SO2 (VolcanEESM) (Carn et al., 2016; Neely III and Schmidt, 2016). We branched out the experiment ensemble with Mt. Pinatubo eruption (GISS-PIN-SO2) and the counterfactual ensemble without Mt. Pinatubo (GISS-NOPIN-SO2) from the historical (GISS-HIST-SO2) using perturbed initial conditions (1st Jan 1986) from the year 1986 to 1999. The perturbation to the initial conditions is generated by altering the radiation-related random number generator that deals with fractional cloudiness in the column.*"

*Table 1: Simulation experiment details.*

| EXP Name | Description | Time period /run length | # of ensembles |
|---|---|---|---|
| GISS-CMIP6-PI | Preindustrial | 1850 climatology /1400 years | 1 |
| GISS-PI | Preindustrial | 1850 climatology /500 years | 1 |
| GISS-HIST-SO$_2$ | historical | 1850-2014/165 years | 1 |
| GISS-PIN-SO$_2$ | historical | 1986-1999/ 15 years | 11* |
| GISS-NOPIN-SO$_2$ | historical | 1986-1999/ 15 years | 11* |

*\*These ensemble members are branched out from the GISS-HIST-SO$_2$ by perturbing the initial conditions.*
"

Line 259: Clarify what is meant by "ground energy." Does "incoming solar radiation" refer to net solar radiation (downward minus upward)?

Thank you for the correction. Rn is the net solar radiation and G is the net ground heating. We modified the text accordingly and we verified that in the code we used the correct variable (line 277-280).

Line 261: What does "0.0864/2.45" represent? Please clarify.

*This factor represents a simplified form of unit conversion form W/m2 to mm/day. We modified it in the equations and added an explanation of it in the text (line 280): "(units are mm per day; 1 Wm$^{-2}$ = 0.0353 mm/day)". Also modified the equations 2.4.2b and 2.4.2c.*

Line 338: Define "MSU."

*Modified and inserted the full form and moved the relevant discussion in the supplement, following the recommendation of the other reviewer (Supplementary text section S3.0; line 75).*

Line 358: Note that TLS refers to "temperature of the lower stratosphere," not "tropical lower stratosphere."

*Modified the sentence and moved the relevant discussion in the supplement, following the recommendation of the other reviewer (Supplementary text section S3.0)..*

Line 369: Consider using the "seasonal variations in incoming solar radiation."

*Modified the sentence (supplementary section S3.0 line 84).*

Line 446: "Then" should be corrected to "than."

*Thanks, corrected (line 439).*

Lines 566-569: Please describe the drivers in the same order as presented in the figure.

*Modified consistently across all figures (line 559-562; 597-600; 643-646).*

Line 615: What is meant by "post-1050"?

*Thanks, corrected as it was a typo. "post-1950"(line 605).*

Lines 694-695: There should be a minus sign before "8 W m$^{-2}$."

*Thanks, This is deleted now as the part of conclusion is modified now..*

**Reference:**

Carn, S. A., Clarisse, L., and Prata, A. J.: Multi-decadal satellite measurements of global volcanic degassing, Journal of Volcanology and Geothermal Research, 311, 99–134, https://doi.org/10.1016/j.jvolgeores.2016.01.002, 2016.

Neely III, R. R. and Schmidt, A.: VolcanEESM: Global volcanic sulphur dioxide (SO2) emissions database from 1850 to present - Version 1.0 (1.0), https://doi.org/10.5285/76EBDC0B-0EED-4F70-B89E-55E606BCD568, 2016.

---

## Author Comment (AC2)

Authors' Response to Reviewer 2

**Review of "Mount Pinatubo's effect on the moisture-based drivers of plant productivity „by Ram Singh et al. 2023**

The aim of the current study is to understand how volcanic eruptions affect ecohydrological conditions and plant productivity. The authors used the NASA Earth System Model to simulate the 1991 eruption of Mt. Pinatubo and to detect its response on soil moisture and evapotranspiration as short-term ecohydrological controls on plant productivity. Using the Soil Moisture Deficit Index (SMDI) and the Evapotranspiration Deficit Index (ETDI), they find that about 10-15% of the land area shows statistically significant dry or wet patterns in the volcanically perturbed climate conditions for 1992 and 1993, and between 5-10% in the following years (1994-1995). The authors focus on three regions that show different responses in EDTI and SMDI. In Equatorial Africa, decreases in both indicate a likely negative impact on crop productivity, while in the Middle East region increases indicate a positive impact on crop productivity. North Asia on the other hand, shows an increase in SMDI and a decrease in ETDI, indicating that crop productivity has probably decreased, but not due to water-related factors.

The paper needs some major improvements. It needs to be more streamlined and the results should be discussed in a broader context, see general comments below. I therefore recommend publication only after major revisions.

We sincerely appreciate the reviewer's thoughtful comments on the manuscript. We have carefully addressed these comments, resulting in substantial revisions that have significantly improved the overall presentation of the results. Responses to specific comments are provided in blue text under each comment, with italicized text indicating specific additions to the manuscript. Pointers to specific changes in the main manuscript are highlighted in red brackets.

General comments

- Part of the paper reads like a model evaluation paper of the climate response to volcanic forcing in the MATRIX version of the GISS ModelE2.1 (Bauer et al., 2020). I am therefore confused as to the purpose of this paper. If validation of the Pinatubo simulation is one of the aims of the paper, this should be clearly communicated. The evaluation of the primary dependent variables temperature and precipitation is quite lengthy and has been done before in other contexts, see points below.

  Indeed, a model evaluation is not the goal of this manuscript, but we have to present some evaluation to demonstrate that the model has skill in simulating the climate impacts of volcanic aerosols. This also highlights the modeling capabilities of GISS ModelE (MATRIX) for interactively simulating the volcanic aerosol properties and climate responses which governs the ecohydrological conditions that further affect the primary productivity.
  To reduce the the length of this section, we decided to move a large portion of the microphysical and radiative properties discussion (including figure 1 in the submitted version is now figure S3 in supplementary info) of volcanic aerosols to the supplementary information (section S2; line 39 in supplementary info).

- Regarding the evaluation, I wonder why you do not put your aerosol microphysical model results into a broader context and relate them to recent work. Quaglia et al (2023) published last year an extensive multi-model data comparison of the Pinatubo episode with different aerosol microphysical models.

  Thanks for pointing this out. Quaglia et al., (2023) has demonstrated the model inter-comparison results for the experiment protocols covering a range of strategies for volcanic aerosol injection strength and altitudes and demonstrated the control of various factors on simulating the microphysical properties of volcanic aerosols in different aerosol microphysics. The estimate of the injected SO2 and plume height for the Pinatubo eruption followed in our study (SO2=15.2 Tg, Height: 25km) are not exactly equal with the cases studied by Quaglia et al., 2023. Their model setup nearest to our simulation uses 7 TgS (14 Tg of SO2) and a plume height of 22 km, and another with the same amount of $SO_2$ and a plume height in the range of 18-25 km. We have added the following lines in the description, which is currently supplementary section S2 (line 52-59 in supp. info), since this part is now in the supplement to present our MATRIX results against the broader context of the 6 models utilized by Quaglia et al., (2023).

  *"Quaglia et al. (2023) have presented a detailed evaluation of the control of aerosol injection strength and altitude on microphysical properties of volcanic aerosols using models with interactive chemistry and microphysics under the Interactive Stratospheric Aerosol Model Intercomparison Project (ISA-MIP). Broadly, ModelE (with the MATRIX aerosol microphysics code) simulated well the evolution of the volcanic plume (AOD, effective radius, and aerosol dispersion) compared to the closest match to our configuration of a sulfur injection strength (~ 7 TgS = 14 Tg of SO2) at injection heights both at 22km and the range 22-25 km presented by Quaglia et al., 2023."*

- The GISS E2.1 model has been used before to study the impact of volcanoes on climate, so this aspect is not really new. I wonder why you do not discuss the climate response after the Pinatubo eruption in your model version with that simulated with the CMIP6 version of the GISS ModelE2.1 (Kelly et al. 2020) in the historical runs and in the Pinatubo VolMIP ensemble (Weierbach et al., 2020). From my point of view, the only open/interesting question here is I how does the surface climate response change when you calculate the aerosol microphysics online.

  - Kelley et al. (2020) and Weierbach et al. (2023) (we think by Weierbach et al. (2020) the reviewer meant Weierbach et al. (2023)) have used a prescribed volcanic forcing as monthly mean AOD in the stratosphere, in line with the CMIP protocol. The volcano response when using monthly mean climatology of AOD can be very different from when $SO_2$ is injected in the atmosphere and chemistry converts it to sulfate, and then dynamics transport it around. MATRIX has been used elsewhere (LeGrande et al., 2016; McGraw et al. 2024, Osipov et al. 2021, Singh et al.2023) for different eruptions, including Pinatubo, but the aim of this manuscript is not to compare against them. Recently, McGraw and Polvani (2024) pointed out the importance of including the interactive treatment of volcanic aerosols in context to better explain the volcanic impact on rainfall, by attributing the direct effect of aerosol-radiation interactions through the stratosphere-tropospheric exchange of energy. This can have further

impacts on clouds and regional rainfall, together with a thermodynamical shift of ITCZ at the surface.

We modified the relevant section (line 112-119) and added the following sentences in the manuscript to highlight the importance of using the interactive aerosol chemistry version of GISS model for this study:

*"We assess the impact of the Mt. Pinatubo eruption on the model-simulated climate via multiple pathways, ranging from primary dependent variables to higher-order responses that influence plant productivity. The use of prognostic aerosols enhances the simulations by capturing dynamically consistent feedbacks between the climate response and volcanic aerosols, including aerosol-radiation interactions and stratosphere-troposphere energy flux exchanges (McGraw and Polvani, 2024)"*

- Related to this. I do not understand why you need an aerosol model for your study of the impact of volcanic forcing on moisture-based drivers of plant productivity. This study would also work with prescribed volcanic forcing: you could use the historical CMIP6 ensemble (Miller et al., 2021) or the 81-member VolMIP Pinatubo ensemble (Weierbach et al., 2023). For these simulations, you would have more than 11 ensembles, which would make your results even more statistically robust.

  > Miller et al., 2021 prescribe the volcanic aerosol using the extinction and aerosol size based upon the model estimate and observations. Weierbach et al., (2023) has also used that version of the GISS model, but importantly the VolMIP experiments are conducted with a pre-industrial (PI) climate.
  >
  > Since this study focuses on the volcanically driven climate pathways of impact assessment, we utilized the interactive chemistry version of the GISS model to include all climate feedbacks to the volcanic eruption-generated radiative perturbation (McGraw and Polvani, 2024). Additionally, the climate metrics that need weekly scale data, cannot be calculated accurately when using prescribed, monthly and zonally mean forcings.

The authors are right that there are not many studies on this recent topic. However, soil moisture changes due to volcanic eruptions have been discussed in the context of volcanic impacts on the carbon cycle (e.g. Fröhlicher et al., 2011). There are also some interesting discussions in Zuo et al. (2019a ,b) on the hydroclimate response after a volcanic eruption, where not soil moisture but other relevant hydroclimate parameters are related to NPP. Furthermore, there is a broad discussion in the geoengineering community about the impact of stratospheric aerosol on soil moisture and food production for solar geoengineering, see for example Cheng et al. (201). These issues should be addressed in the paper.

We appreciate the reviewer for pointing out these studies. We have included the following sentences addressing the relevance of these studies in context to the objectives of this manuscript.

*"Studying a large (10xPinatubo) volcanic eruption, Fröhlicher et al. (2011) have shown that the terrestrial carbon pool is sensitive to the regional (in the tropics and sub-tropics) soil-moisture content through the net-ecosystem productivity. Using the geoengineering large ensemble simulations with CESM model, Cheng et al. (2019) have analysed the changes in terrestrial hydrological cycle and discussed the future soil-moisture response and its drivers under a geoengineering scenario"*

*"It is also shown that volcanic eruptions can alter regional rainfall and hydroclimate in general, which could prominently affect regional plant productivity (Zuo et al., 2019a;b)."*

- The authors speculate a lot in the paper about potential impacts on crop productivity, but they have not shown any GPP or NPP anomaly plots. I wonder why, as this would strengthen the paper considerably.

  Thank you for pointing this out. The aim of this study is to focus on hydroclimate metrics. Transpiration is the most dominant process contributing to AET on land and is strongly correlated with photosynthesis. Thus, an increase in AET serves as a reliable indicator of an increase in GPP. Consequently, we have chosen not to emphasize GPP in this analysis. However, we have revised portions of the manuscript to address this concern. Additionally, we included a plot illustrating the seasonal anomaly of GPP in the supplementary information (Figure S9) and provided a discussion on plant productivity in the conclusions section, along with examples of similar findings from other studies (line 712-728; Conclusion section).

*"Kandlbauer et al., (2013) examined crop responses (using C3 and C4 grasses as proxies) to the 1815 Tambora eruption using the HadGEM-ES model in three regions very similar to those in our study. Their findings suggest that plant productivity decreases with positive changes in soil moisture in the higher-latitude Asian region. In the mid-latitudes over the Southern Europe/Middle East region (adjacent to our MDE region), volcanic eruptions may enhance plant productivity by providing additional soil moisture through increased rainfall. However, in the MDE region in our study, we found that the applied irrigation also benefits soil moisture supply along with the increased rainfall. Furthermore, both studies report a decrease in productivity in the tropical region. In general, these results complement the findings of this study, which suggest that if sufficient water is available in the Southern Europe/Middle East region, volcanic eruptions may enhance plant productivity. In contrast, in the far northern latitudes, water is not the primary driver of plant responses, and productivity is likely to decline. Seasonal-scale changes in gross primary productivity (GPP) confirm the regional trends in plant productivity following the eruption. The simulations show a more pronounced decrease in GPP in the northern high-latitude*

*region and a significant increase in GPP over the European and Mediterranean regions. Additionally, distinct patterns of decrease and increase in GPP are simulated in the tropical northern and southern regions, respectively (Figure S9)).*"

Specific comments

- Specific comments
-

| | |
|---|---|
| Line 23-24: | Could be deleted. if model evaluation is not a specific subject of your paper. Modified the sentence by removing the radiative response. (line 23-24) |
| | *"The model simulates a mean surface cooling of ~0.5 °C following the Mt. Pinatubo eruption."* |
| Line 27: | You do not show any agricultural response in the paper so please be careful with your wording. Here, Agricultural response points to the agricultural drought indices (SMDI & ETDI), we have re-written the sentence to clarify this: |
| | (line 26-28) *"We find that up to 10-15% of land regions show a statistically significant hydroclimate response (wet and dry) as calculated by the Soil Moisture Deficit Index (SMDI) and Evapotranspiration Deficit Index (ETDI)."* |
| Line 28: | Not clear what you mean with "these higher-order impacts". Re-written the sentence (line 28-29) as |
| | *"Results confirm that these impact metrics successfully present a more robust understanding of plant productivity"* |
| Lines 41-44 | Too many references for something well known and obvious. I suggest that you refer here instead to some overview papers e.g. Marshall et al (2022), Kremser et al (2016), Timmreck et al (2012). Modified and added new references (line 41-43) |
| Line 48: | Again, you can reduce the amount of citations and refer to some overview papers or the recent model intercomparison paper. Modified and citation added (line 47) |
| Line 53 ff: | See point above. Modified (line 53) |
| Lines 73 ff: | Please cite here in addition or instead of Toohey et al (2016) the most recent paper for the mid-6th century volcanic impact on post-volcanic climatic and societal response over Scandinavia: van Dijk et al (2023). Thanks, citation included at line 70-71 |
| Lines 193-195: | It is not clear here why you use the climatology of the years 1950-2014, maybe you refer here already to the supplementary material. 1950-2014 climatology is used to calculate the anomalies generated due to volcanic perturbation of Pinatubo eruption. Also, this period serves as the |

reference calibration period for agricultural drought indices (SMDI & ETDI) calculation. Please also check our response to the other reviewer about PCH, and the new figure S2. Relevant explanation in main text modified in line 207-218.

"…….However, directly comparing the difference between the two ensembles (PCH and NP) is an alternative approach to presenting the Pinatubo effect (see Supplement Figure S2). Using either approach leads to the same general conclusions, with only small quantitative differences. Nevertheless, we chose to remain consistent with the baseline requirements for other metrics as well and used the historical climatology for period 1950-2014 as the baseline for the core of our analysis. The coloring emphasizes the significant regions of anomalies. But we also emphasize the difference in calculations: the grey areas show no significant change between the PCH and NP ensembles, while the anomalies are PCH ensemble mean minus climatology"

| | |
|---|---|
| Line, 227: | Index instead of Indices |
| | Thanks, Corrected |

Lines 239-240:  Please use SI units, even its not typical for SDI, you can put the "feet metric" in brackets
Since the soil-moisture indices nomenclature has distinguished using the depth in feet, we prefer to leave it in feet. We also added the equivalent numbers in SI unit in brackets throughout, as additional information.

Line 247ff:  Here and in the following lines the reference style seems not correct.
Corrected

Lines 284-286:  Concerning the justification, it is quite difficult to see in the supplementary Fig S1 the difference between the three reference periods. Maybe a difference plots between them would more useful.
Figure S1 shows the anomaly for the year 1992 with respect to 3 different reference climatology periods. The difference of the differences can be more confusing than explanatory.

Lines 309-311:  This sentence sounds strange, please reformulate.
Modified as follows: Line 317-320 "...which then decreases with time due to the deposition of volcanic aerosols (English et al., 2013; Sato et al., 1993). In this study, the model-simulated aerosol optical depth (AOD) due to volcanic aerosol and radiative forcing is larger than the previously reported AOD of 0.15 and forcing of -4.0 to -5.0 $Wm^{-2}$ due to the Mt. Pinatubo eruption (Hansen et al., 1992; Lacis et al., 1992)"

Lines 357-370:  It is not clear to me why the lower stratospheric temperature response is relevant for your specific topic
We modified the Figure 2 (submitted version) as Figure 1 (revised version) and removed the panel showing the lower stratospheric temperature response. We also modified the result discussion and moved it to the supplementary text (section S3.0), along with figure S4.

Lines 416 -421:  So why choose for your analysis few realizations with prognostic aerosol

instead of more realizations with many members?
We used all 11 realizations (which is not a few, in our opinion) for the analysis. Here, we are pointing out the possible causes of higher and spatially varying characteristics of rainfall. Polvani et al. (2019) pointed out that a large ensemble is good for the robust response, but it is computationally expensive. Singh and AchutaRao (2019) showed that our set of 11 ensemble is sufficient to represent significance in climate response at the regional scale.

We added the following in the text and modified the explanations (Line 407-415):

*"We acknowledge the signals due to the model's internal variability when averaging the impacts across multiple ensembles, but 11 ensembles are a good compromise between few vs. many ensemble members which was shown to be sufficient to represent significance in climate response at the regional scale (Polvani et al., 2019; Singh and AchutaRao 2019)."*

Line 436:  Section 3.4.
Thanks, corrected (Line 430)

Lines 444-446:  This sentence is not clear, please reformulate.
Modified (line 438-439)

Line 535:  What are "elements of a time lag between precipitation"?
We modified the sentence (line 528-530):

*"Additionally, SMDI_2 (top 2 feet or 0.6 m) and ETDI have demonstrated a slow development of drought conditions, beginning by the end of the year 1991 (SON season), reflecting a time lag between seasonal precipitation patterns (Narasimhan and Srinivasan 2005)."*

Lines 644 ff:  Does this happen in all realizations or is it just a coincidence in the ensemble mean? How meaningful are changes of individual weeks?
We highlighted these 2 weeks as simulated in the ensemble means response (shown in Figure 9). These sentences (line 635-636) shows the importance of considering such high temporal resolution impact metrics in context to agricultural productivity regardless of volcanic forcings.

Lines, 691 ff:  I think you can shorten the part about the model performance/evaluation substantially. You need not to list the references here when you already have included them in the text
Thanks for pointing this out, we modified the text (entire paragraph between lines 680-702 and kept the only lines 683-685) and deleted the earlier cited references.

Line 719:  "volcanic forcings due to the Mt. Pinatubo eruption" sounds strange, please Revise
Modified the full sentence as follows (line 689-692):

*"These drought indices confirm the moisture-driven dry and wet patterns*

*observed in early 1992 and the following years over the tropical regions and mid-latitudes of the Northern Hemisphere, respectively, as a response to the radiative perturbation caused by the Mt. Pinatubo eruption."*

- Tables:
  - Table 1 can be merged with the figure caption of Figure 8

    We think the reviewer refers to table 2 here. We have deleted this and the details of the regions are now included in the caption of figure 7, as suggested (line 553).

- Figures:
  - In general: The multi-panel figures (5, 6,7) are too small and hard to read and therefore not very convincing. I strongly recommend to reduce the number of panels by either showing only specific years or specific seasons. This would make the figures much more readable and therefore much better emphasize the point. The missing panels (seasons, years) could be put into the supplementary material.

    -Thanks for the suggestion. We have modified these figures (Figure 4,5,6 in main text, and Figure S6 in supplementary) by zooming over each panel by removing the high latitude regions over the Arctic and Antarctica where no vegetation exists.

  - Figures 3, 4 : 1st two panels are useless and could be deleted.
    - Thank you for pointing this out, We preferred to keep the DJF and MAM as a confirmation of our methodology which shows that the mean climate under both experiments (Pinatubo & counterfactual case) are same till the eruption in June 1991.
    - References:
      - Please update the reference to Brown, H. Y, Geosci. Model Dev., 17, 5087–5121, https://doi.org/10.5194/gmd-17-5087-2024, 2024.

    Modified

References

Bauer, S. E., Tsigaridis, K., Faluvegi, G., Kelley, M., Lo, K. K., Miller, R. L., Nazarenko, L., Schmidt,G. A., and Wu, J.: Historical (1850–2014) Aerosol EvoluDon and Role on Climate Forcing Using the GISS ModelE2.1 ContribuDon to CMIP6, Journal of Advances in Modeling Earth Systems, 12, e2019MS001978, h\ps://doi.org/10.1029/2019MS001978, 2020.

Cheng, W et al.  Soil moisture and other hydrological changes in a stratospheric aerosol geoengineering large ensemble. Journal of Geophysical Research: Atmospheres, 124, 12,773–12,793 https://doi.org/10.1029/2018JD030237, 2019.

English, J. M., Toon, O. B., and Mills, M. J.: Microphysical simulations of large volcanic eruptions: Pinatubo and Toba, Journal of Geophysical Research: Atmospheres, 118, 1880–1895, https://doi.org/10.1002/jgrd.50196, 2013.

Frölicher, T. L., Joos, F., and Raible, C. C.: Sensitivity of atmospheric $CO_2$ and climate to explosive volcanic eruptions, Biogeosciences, 8, 2317–2339, https://doi.org/10.5194/bg-8-2317-2011, 2011.

Hansen, J., Lacis, A., Ruedy, R., and Sato, M.: Potential climate impact of Mount Pinatubo eruption, Geophysical Research Letters, 19, 215–218, https://doi.org/10.1029/91GL02788, 1992.

Kelley, M. et al.: GISS-E2.1: ConfiguraDons and Climatology, Journal of Advances in Modeling Earth Systems, 12, e2019MS002025, https://doi.org/10.1029/2019MS002025, 2020.

Kremser, S., et al.: Stratospheric aerosol – Observations, processes, and impact on climate, Rev. Geophys., 54, 1–58, https://doi.org/10.1002/2015RG000511, 2016.

Lacis, A., Hansen, J., and Sato, M.: Climate forcing by stratospheric aerosols, Geophysical Research Letters, 19, 1607–1610, https://doi.org/10.1029/92GL01620, 1992.

Marshall, L, E C. Maters, A. Schmidt, C. Timmreck, A. Robock, and M. Toohey, Volcanic effects on climate: recent advances and future avenues. Bull Volcanol 84, 54. https://doi.org/10.1007/s00445-022-01559-3, 2022.

McGraw, Z., DallaSanta, K., Polvani, L. M., Tsigaridis, K., Orbe, C., and Bauer, S. E.: Severe Global Cooling After Volcanic Super-Eruptions? The Answer Hinges on Unknown Aerosol Size, Journal of Climate, 37, 1449–1464, https://doi.org/10.1175/JCLI-D-23-0116.1, 2024.

Miller, R.et al.: CMIP6 historical simulations (1850–2014) withGISS-E2.1, J. Adv. Model. Earth Syst., 13, e2019MS002034, https://doi.org/10.1029/2019MS002034, 2021

Narasimhan, B. and Srinivasan, R.: Development and evaluation of Soil Moisture Deficit Index (SMDI) and Evapotranspiration Deficit Index (ETDI) for agricultural drought monitoring, Agricultural and Forest Meteorology, 133, 69–88, https://doi.org/10.1016/j.agrformet.2005.07.012, 2005.

Quaglia, I., Timmreck, C., Niemeier, U., Visioni, D., Pitari, G., Brodowsky, C., Brühl, C., Dhomse, S. S., Franke, H., Laakso, A., Mann, G. W., Rozanov, E., and Sukhodolov, T.: Interactive stratospheric aerosol models' response to different amounts and altitudes of SO2 injection during the 1991 Pinatubo eruption, Atmos. Chem. Phys., 23, 921–948, https://doi.org/10.5194/acp-23-921-2023, 2023.

Sato, M., Hansen, J. E., McCormick, M. P., and Pollack, J. B.: Stratospheric aerosol optical depths, 1850–1990, Journal of Geophysical Research: Atmospheres, 98, 22987–22994, https://doi.org/10.1029/93JD02553, 1993.

Singh, R. and AchutaRao, K.: Quantifying uncertainty in twenty-first century climate change over India, Clim Dyn, 52, 3905–3928, https://doi.org/10.1007/s00382-018-4361-6, 2019.

Timmreck C. (2012), Modeling the climatic effects of volcanic eruptions, Wiley Interdisciplinary Reviews: Climate Change doi: 10.1002/wcc.192.

Toohey, M., Krüger, K., Sigl, M., Stordal, F., and Svensen, H.: ClimaDc and societal impacts of a volcanic double event at the dawn of the Middle Ages, ClimaDc Change, 136, 401–412, https://doi.org/10.1007/s10584-016-1648-7, 2016.

van Dijk, E., I. Mørkestøl Gundersen, A.de Bode, H. Høeg, K. Loftsgarden, F. Iversen, C. Timmreck, J. Jungclaus and K. Krüger (2023). Climatic and societal impacts in Scandinavia following the 536 and 540 CE volcanic double event, Clim. Past, 19, 357–398, https://doi.org/10.5194/cp-19-357-2023, 2023.

Weierbach, H., LeGrande, A. N., and Tsigaridis, K.: The impact of ENSO and NAO initial conditions and anomalies on the modeled response to Pinatubo-sized volcanic forcing, Atmos. Chem. Phys., 23, 15491–15505, https://doi.org/10.5194/acp- 23-15491-2023, 2023.

Zanchettin, D et al.: Effects of forcing differences and initial conditions on inter-model agreement in the VolMIP volc- pinatubo-full experiment, Geosci. Model Dev., 15, 2265–2292, https://doi.org/10.5194/gmd-15-2265-2022, 2022.

Zuo, M., Zhou, T., and Man, W.: Hydroclimate Responses over Global Monsoon Regions Following Volcanic Eruptions at Different Latitudes, J. Climate, 32, 4367–4385, https://doi.org/10.1175/jcli-d-18-0707.1, 2019a

Zuo, M., Zhou, T., and Man, W.: Wetter Global Arid Regions Driven by Volcanic Eruptions, J. Geophys. Res.-Atmos., 124, 13648–13662, https://doi.org/10.1029/2019jd031171, 2019b.

---

## Author Response (AR1)

**Mount Pinatubo's effect on the moisture-based drivers of plant productivity**

Ram Singh1,2, Kostas Tsigaridis1,2, Diana Bull3, Laura P Swiler3, Benjamin M Wagman3, Kate Marvel2

**Response to the comments-**

Dear Editor,

Thank you for facilitating the review process for the manuscript *egusphere-2024-2280*. We have addressed all comments from both reviewers and uploaded our responses. During this process, the manuscript has been substantially revised, resulting in an improved presentation of the results.

Regarding the comments from both Reviewer 1 and Reviewer 2, we have carefully addressed each point and incorporated the corresponding changes into the main manuscript and supplementary information. Responses to each comment are provided in blue text, with italicized text indicating specific additions to the manuscript and supplementary materials. Pointers to the corresponding changes in the main text are highlighted in red text. Both response files are uploaded at the designated links in the review/discussion section at the manuscript pre-print.

We believe we have adequately addressed all the comments and concerns. Please let us know if there are additional issues or areas we may have overlooked. We look forward to hearing about the next stage of the review process.

**Sincerely,**

Ram Singh and co-authors

---

## Referee Report (RR1)

This manuscript aims to study the effect of volcanic eruptions on moisture-based drivers of plant productivity. The results indicate a 0.5 °C surface cooling due to the Mt. Pinatubo eruption. However, this is not a major finding, as previous studies have already reported this level of surface cooling following the volcanic activity at Mt. Pinatubo.

The major flaw of the study is the lack of a direct statistical analysis of the impact of atmospheric air pollution from volcanic activity on climatic and hydroclimatic parameters. Individual anomalies in simulated explanatory variables were inferred as results of volcanic eruptions, but only over 10-15% of the area. No direct evidence of the decrease or increase in plant productivity is presented. Terminology in the manuscript is not clearly defined. Section 2.3 on methods is difficult to understand and poorly written. The interchangeable use of terminologies adds to the confusion. For example, "counterfactual (sometimes counter-factual) ensemble simulation" and "no-Pinatubo ensemble simulation" refer to the same simulation but are used interchangeably throughout the text.

**Please find below some additional comments:**

In Figure S3, abbreviations such as LW, WS, and NET are not defined either in the main text or in the figure caption.

Line (337). "The zonal AOD shows the dispersion and transport of aerosol poleward after the eruption." However, it is not clear from Figure 1 how this conclusion is achieved. It would be better if the grid cell level variation with temporal scale is represented after the eruption event.

Line343: QBO is not defined.

Line 343-355: The spatial connection of AOD's effect on temperature changes cannot be established using Figure 1. Grid-level correlations could provide more insight into their connectivity.

The seasonal anomalies of temperature and rainfall presented in Figure 2 and Figure 3 are with reference to long-term climate conditions that is from 1950-2014. No rational is provided for this approach either in the result section or in method section. How to account for GHGs radiative effect during this long-term reference period? The anomaly comparison years for

temperature are 1991-1995 but for rainfall only two years are presented. No explanation for this inconsistency is available.

Lines 335-335: The results shown in Figure 4 are significant only over forest land (Congo tropical forest and Russian boreal forest), which has a much deeper root zone compared to croplands. In Figure S6, the land mass area has pixels of a similar color to the ocean. Is there any explanation for this?

Since the study is focused on drivers of plant productivity, remove the ocean region from temperature and rainfall figures to have consistent study area with SMDI and ETDI.

Figure 7 should just be in supplementary materials. However, the rationale for selecting areas in different regions is unclear. It appears that the selection is based on regions with clusters of grid cells showing significant anomalies.

A point-by-point response to the first review was not provided by the author, making it difficult to evaluate whether the major concerns were fully addressed. However, upon reviewing the track-changes-enabled draft, the following anomalies were observed.

Lines (172-177): Why delete all the lines when only change in the original text is the reference to Figure and correction in cited paper?

It's not clear what revision is made between lines 260-265. It seems like cut and paste.

Lines between 378-384, All the revision is cut-paste only change is reference to Suppl. Figure.

Revision between Lines 763-775 appears to be selecting alternative word. It does not add value to the text.

---

## Author Response (AR2)

**Editor:**

**Dear editor,**

You may find below the replies to the three reviewers of the second round of review, together with some replies to your comments. We believe that we have addressed all reviewers' comments, and tried to stress cases where we had already replied to past rounds of reviews sufficiently, which was missed in the second round of reviews. A clear example is reviewer #4 who says that we have not provided point-by-point replies to the first round of reviews, which is obviously not the case.

The fifth reviewer has recommended a rejection, and raised concerns, like the previous three reviewers, about the methodological approach and assumptions behind it.

Thank you for giving us the opportunity to reply. Based on what we see in the manuscript submission page, two reviewers said major corrections (but no rejection), and our replies to them are already online since the last review round. One reviewer said minor corrections and accept, and two reviewers said reject (not three). For the latter three reviews, you can find our detailed replies below. We are also submitting a track-change version of the manuscript, as required by the journal.

All reviews, regardless of their recommendation on acceptance or rejection, were constructive, and helped to greatly improve the manuscript. They asked for explanations of methodology, a restructuring of the results section, and the inclusion of some new results to further support our conclusions. We performed all such requested items.

We would like to highlight a few cases where we respectfully disagreed with the reviewers: reviewer #4 asked for a statistical analysis of air pollution, but we never claimed that we are doing anything related with air pollution in the manuscript; reviewer #5 denied that soil moisture is a proper driver of agricultural productivity, and we made a case on why this is, in our opinion, an incorrect claim; and both reviewers #4 and #5 said that we are not presenting an analysis on model response for GPP/NPP, which in fact is included in the revised manuscript from the first round (the one that both reviewers were reviewing) as a response to the first round of reviews.

We invite you to pay particular attention to the weight of the comments of reviewer #4, which, in our opinion, are not major enough to justify a straight-out rejection. Also worth checking closely is our reply to the last comment of reviewer #5, which summarizes all the changes we have done over the two review rounds.

While we acknowledge that the authors have made an effort to address the comments from the first round of reviews, the revised results are not sufficient to convince independent referees of the value of the study. There seems to be a fundamental lack of understanding and/or of convincingly illustrating the purpose and methodology of using the drought indices to derive the presented conclusions.

Please see the replies to the first comment of reviewer #5, which outlines why we believe our methodology is, indeed, scientifically sound.

The authors are given the chance to respond to the second round of reviews (#3 through #5), but are reminded that the outcome of a further revision will not necessarily lead to acceptance despite the one positive review, and that as the fifth reviewer also declined to re-review any revised manuscript, any resubmission will likely experience another delay in the manuscript processing due to difficulty in finding more reviewers.

Thank you once more for the opportunity given. We understand that the decision might take longer, given your statement that of the five reviewers one is not willing to rereview the manuscript. We are looking forward to your decision.

Responses to each comment are provided in blue text, with italicized text indicating specific additions to the manuscript and supplementary materials.

**Reviewer 3:**

Responses to each comment are provided in blue text, with italicized text indicating specific additions to the manuscript and supplementary materials.

Overall this work is very good organized, and the model design are very reasonable. The paragraph figure structure is very reader friendly. So I recommend a technical corrections, for the minor technical issues.

Thank you.

line 164 CH4: use subscription

Done.

line 264-267: this sentence is too long and hard to understand, since PDSI and CMI has been quoted before, you can just use this abbreviation without mentioning the full name or the citation. Even though, this sentence is also too long, please consider to divide it into two.

We modified the sentence:

"The limitations of PDSI and CMI in the formulation used for the PET calculation (Thornthwaite, 1948), together with the lack of consideration of land cover types on the water balance, have encouraged the exploration of ETDI for agricultural productivity."

line 352-354: is there a missing 'is' before consistent? Yes, fixed.

**Reviewer 4:**

Responses to each comment are provided in blue text, with italicized text indicating specific additions to the manuscript and supplementary materials.

This manuscript aims to study the effect of volcanic eruptions on moisture-based drivers of plant productivity. The results indicate a 0.5 °C surface cooling due to the Mt. Pinatubo eruption. However, this is not a major finding, as previous studies have already reported this level of surface cooling following the volcanic activity at Mt. Pinatubo.

Our study does not present the surface temperature response as a novel contribution. Rather, as mentioned as early as in the abstract (quoted below), the primary focus of this work lies in analyzing ecohydrological conditions and their implications. More specifically we focus on drought indices, plant productivity, and their spatiotemporal relevance from seasonal to sub-seasonal (weekly) time scales. Notably, the weekly time-scale analysis provides important insights into agricultural productivity. To the best of our knowledge, this is the first study to offer a comprehensive examination of these critical drivers of plant productivity. The inclusion of surface temperature is intended to support the evaluation of the model's performance and to provide necessary context for the ecohydrological analysis, and the agreement of our model with past studies demonstrates that the temperature response is well within the bounds of past literature.

"Here, we will explore the understudied store (soil moisture) and flux (evapotranspiration) of water as the short-term ecohydrological control over plant productivity in response to the 1991 eruption of Mt. Pinatubo."

The major flaw of the study is the lack of a direct statistical analysis of the impact of atmospheric air pollution from volcanic activity on climatic and hydroclimatic parameters.

The focus here is on the well-established climatic impacts of stratospheric aerosol injections following explosive volcanic eruptions, which perturb Earth's radiative balance and influence climate and hydroclimatic variables. Air pollution is not a topic we study in this manuscript, so trying to correlate air pollution metrics with climatic or hydroclimatic parameters would be out of scope.

About statistics in general, our analysis examines the climatic response to a volcanically perturbed atmospheric radiative balance, extending beyond surface temperature to include higher order impact metrics such as rainfall, soil moisture, and surface heat fluxes (actual and potential evapotranspiration), which are critical to understanding plant productivity through land—atmosphere interactions. The response is detected using a counterfactual inference approach and evaluated through paired Student's t-tests at a 95% confidence level. Responses not meeting this significance threshold are masked, as already mentioned in several of our figures. We present the full suite of responses—including temperature, rainfall, and higher-order drivers—alongside drought indices (SMDI and ETDI), which reflect soil moisture anomalies and atmospheric moisture demand, respectively, and only where statistical significance is calculated.

Individual anomalies in simulated explanatory variables were inferred as results of volcanic eruptions, but only over 10- 15% of the area. No direct evidence of the decrease or increase in plant productivity is presented.

Direct evidence of plant productivity is inferred through soil moisture—based drivers at both global and regional scales, with the geographic dominance of specific drivers discussed in the conclusion. Additionally, we present changes in gross primary productivity (GPP) as direct evidence in the conclusion (lines 751–763 TC/710-726) and in Supplementary Figure S9. This has also been clarified in response to Reviewer 2.

"In contrast, in the far northern latitudes, water is not the primary driver of plant responses, and productivity is likely to decline. Seasonal-scale changes in gross primary productivity (GPP) confirm the regional trends in plant productivity following the eruption. The simulations show a more pronounced decrease in GPP in the northern high-latitude region and a significant increase in GPP over the European and Mediterranean regions. Additionally, distinct patterns of decrease and increase in GPP are simulated in the tropical northern and southern regions, respectively (Figure S10)."

The estimate of 10–15% of global land area represents a substantial and societally relevant extent.

Terminology in the manuscript is not clearly defined. Section 2.3 on methods is difficult to understand and poorly written.

We modified the text of section 2.3 (quoted below), to clarify the message we were trying to pass.

"

2.3 Methods: This study investigates the impacts of the Mt. Pinatubo eruption on the major drivers of primary productivity focusing on soil moisture and evapotranspiration related metrics. We followed the counterfactual inference approach to draw causal inference of the Pinatubo eruption. Hereafter, we use 'PCH' (Mt. Pinatubo and Cerro Hudson) to refer to the 'GISS-PIN-SO2' and 'NP' for the 'GISS-NOPIN-SO2' ensembles. We have included the Cerro Hudson eruption in both ensembles since we are focusing on the Mt. Pinatubo-driven climate response.

2.3.1 Statistical analysis for detecting Mt. Pinatubo-significant regions and anomalies calculations.

We treat the no-Pinatubo ensemble (NP) as a counterfactual climate simulation and utilize it to perform the paired Student's t-test for causal inference. The null hypothesis is that the ensemble means of a quantity of interest (QoI) in a region over a time period are the same between ensembles (i.e. Ho: Ho: Hor). In the subsequent figures in this document, gray regions indicate acceptance of the null hypothesis at the 95% confidence level, while the coloring emphasizes the rejection of the null hypothesis and the significant regions of anomalies relative to 1950-2014 climatology (see Supplementary information section S1.0). However, we also explored the alternate approach of directly comparing the difference between the two ensembles (PCH and NP) for presenting the Pinatubo effect (see Supplement Figure S2). It is concluded that both of the approaches led to the same general conclusions, with only small quantitative differences. Nevertheless, we chose to remain consistent with the baseline requirements for other metrics, and used the historical climatology for the same period 1950-2014 as the baseline for the core of our analysis. Thus, the grey areas indicate no significant differences between the PCH and NP ensembles, while the colored regions represent the statistically significant anomalies, calculated as PCH ensemble mean minus climatology.

"

The interchangeable use of terminologies adds to the confusion. For example, "counterfactual (sometimes counter-factual) ensemble simulation" and "no-Pinatubo ensemble simulation" refer to the same simulation but are used interchangeably throughout the text.

We apologize for the inconsistent terminology. Since our no-Pinatubo simulation (GISS-NOPIN-SO2) is named "NP" in the manuscript, as mentioned in line 199, we removed

the word "counterfactual" from the instances where we were referring to the model simulations and replaced it with "NP".

Please find below some additional comments:

In Figure S3, abbreviations such as LW, WS, and NET are not defined either in the main text or in the figure caption.

We added explanations in captions on what these abbreviations mean.

"Monthly anomaly of longwave (LW), shortwave (SW), and net (NET) radiative forcing simulated by the GISS ModelE for Mt. Pinatubo (PCH) and without Pinatubo (NP) ensembles"

Line (337). "The zonal AOD shows the dispersion and transport of aerosol poleward after the eruption." However, it is not clear from Figure 1 how this conclusion is achieved. It would be better if the grid cell level variation with temporal scale is represented after the eruption event.

We do not exactly understand the reviewer's comment here. Figure 1 clearly shows an increase in zonally-averaged AOD poleward with time, indicating aerosol transport toward higher latitudes. The initial high AOD blob near the volcano (in the tropics during the second half of 1991) migrates both north and south, and about a half year later significant amounts of it have reached both poles. About 3 years after the eruption the whole atmosphere (in terms of AOD load) has returned to its unperturbed state. All these are clearly visible in the figure and the past literature (e.g., Aquila et al., 2012; Anderson et al., 2015; Singh et al., 2023). Presenting grid-cell level variations over five years (60 monthly-mean differences) is challenging and unlikely to add any meaningful insight, since we are not interested in the longitudinal variability, which is either way very minimal.

A plot similar to what the reviewer has asked for would look like the one below, which shows the seasonal structure of grid point AOD for 5 years.

Line343: QBO is not defined.

**Fixed.**

"Meanwhile, the phases of QBO (Quasi-Biennial Oscillation) and local heating also play a crucial role in the poleward and vertical dispersion of stratospheric aerosols"

Line 343-355: The spatial connection of AOD's effect on temperature changes cannot be established using Figure 1. Grid-level correlations could provide more insight into their connectivity.

The correlation of temperature response with AOD is not a direct one and not even a perfect one, as seen in Figure 1. Clouds respond to AOD changes as well, even if the aerosol layer is in the stratosphere, and these cloud changes also impact temperature. Other than the complexity of generating the figure requested, which was already addressed in comment about line 337 above, its interpretation would have been a complex task that would have led us outside the scope of this work. The zonal mean plot presented shows that in general higher AOD and lower temperature correlate, but not exactly, and this is a common finding across modeling studies of the same topic. We decided to make no changes.

The seasonal anomalies of temperature and rainfall presented in Figure 2 and Figure 3 are with reference to long-term climate conditions that is from 1950-2014. No rational is

provided for this approach either in the result section or in method section. How to account for GHGs radiative effect during this long-term reference period?

The explanation for selecting the reference period 1950–2014 is already provided in lines 207–217 (Quoted below), as well as in Supplementary Section S1 and Figure S1. These illustrate the suitability of this period for capturing the volcanic response with a few tests we performed and explain in the text and supplement. We also addressed this in our responses to both Reviewers 1 and 2.

"It is concluded that both of the approaches led to the same general conclusions, with only small quantitative differences. Nevertheless, we chose to remain consistent with the baseline requirements for other metrics, and used the historical climatology for the same period 1950-2014 as the baseline for the core of our analysis"

The anomaly comparison years for temperature are 1991-1995 but for rainfall only two years are presented. No explanation for this inconsistency is available.

The two-year rainfall anomaly comparison was included in order to illustrate the complexity and uncertainty of rainfall response, particularly over land, which motivated us to largely focus on drought metrics instead. The rationale for this is already explained in section 3.3.

Lines 335-335: The results shown in Figure 4 are significant only over forest land (Congo tropical forest and Russian boreal forest), which has a much deeper root zone compared to croplands. In Figure S6, the land mass area has pixels of a similar color to the ocean. Is there any explanation for this?

We assume this is about Flgure 4 and not line 335 which is not relevant. In Figure 6, the lighter grey shading over land compared to the ocean reflects lower soil moisture levels at a depth of 2–4 feet, due to the presence of non-permeable soil layers. This is explained in text section 3.4.1 (paragraph starting at line 505 and 515 in track changes) and figure captions. We have added that explanation to the Figure 5 and Flgure S6 legend.

"The light grey colored regions represent regions of impermeability."

Since the study is focused on drivers of plant productivity, remove the ocean region from temperature and rainfall figures to have a consistent study area with SMDI and ETDI.

Temperature and rainfall are not solely influenced by land surface processes. Masking ocean regions in surface temperature and precipitation analyses is unconventional and results in the loss of critical information, particularly given the strong land—ocean feedbacks. In contrast, SMDI and ETDI are land-specific indicators that capture the direct influence of land surface conditions. As higher-order impact metrics, they are inherently more representative of land-region responses.

Figure 7 should just be in supplementary materials. However, the rationale for selecting areas in different regions is unclear. It appears that the selection is based on regions with clusters of grid cells showing significant anomalies.

The reviewer is correct that the selection was guided by significant regional anomalies, in order to deeply understand the reason why these regions stand out. The rationale for this analysis is already stated in the abstract and supported by region-specific conclusions (see e.g. Abstract line 29-31, section 3.6 (quoted), paragraph Conclusion section). However, we do agree that this figure can move to the supplement, which is now the new Figure S7 there.

"Considering the complexity of the representation of spatial features, we selected three distinct regions (shown in Figure S7 and detailed in caption) in the northern hemisphere based on the climate response to Mt. Pinatubo in the seasonal analyses presented in Section 3.0"

A point-by-point response to the first review was not provided by the author, making it difficult to evaluate whether the major concerns were fully addressed.

The point-by-point responses to both reviewers are publicly available on the journal discussion page (see <a href="https://doi.org/10.5194/egusphere-2024-2280-AC1">https://doi.org/10.5194/egusphere-2024-2280-AC1</a> and <a href="https://doi.org/10.5194/egusphere-2024-2280-AC2">https://doi.org/10.5194/egusphere-2024-2280-AC2</a>). The reviewer might have missed the binder icon to the right, which links to our reply.

However, upon reviewing the track-changes-enabled draft, the following anomalies were observed.

Lines (172-177): Why delete all the lines when only change in the original text is the reference to Figure and correction in cited paper?

Apologies for this, we are not sure why the whole sentence appears modified. Indeed we only modified the figure reference and a citation that was incorrect.

It's not clear what revision is made between lines 260-265. It seems like cut and paste.

Apologies for this, we are not sure why the whole sentence appears modified. The only change is the word "to" that became "of".

Lines between 378-384, All the revision is cut-paste only change is reference to Suppl. Figure.

One more case where for reasons unclear to us the word processor decided that the whole sentence was modified. Apologies.

Revision between Lines 763-775 appears to be selecting alternative word. It does not add value to the text.

This is correct, since one of the comments we received was to improve the text, and this is exactly what we tried to do. As the present reviewer is not recommending any specific changes, we assume that their comment is a statement, rather than a request for further changes.

**Reviewer 5:**

Responses to each comment are provided in blue text, with italicized text indicating specific additions to the manuscript and supplementary materials.

Currently, this manuscript sits in a weird spot. Methodologically, it didn't to validate the simulated atmospheric response, which is understandable. But the critical problem lies in analysis.

"Moisture-based drivers of plant productivity" is a bad middle ground between hydroglogy and ecophysiology - They could have directly look at GPP/NPP, and examine how temperature/hydrological effects drive those changes; alternatively they could focus on hydrology, which could have broader implications beyond terrestrial ecosystem productivity (e.g. flood risks). Instead, they focus on 2 drought indices (that doesn't offer much extra insight) and made unbacked speculations about how that would impact terrestrial ecosystem/agriculture. From the record of the last round of revision, it doesn't seem that the authors are aware of this major issue.

We respectfully disagree with the reviewer's comment regarding soil moisture as an unsuitable indicator for linking the hydrological cycle and plant productivity. Soil moisture is a key hydrometeorological variable that directly influences plant health particularly in agricultural systems - by supporting photosynthesis, nutrient transport, and root development (Seneviratne et al., 2010; Dirmeyer et al., 2006; Munoth et al., 2006; Garg et al., 2016). As shown in this study, the precipitation signals across timescales possess a high uncertainty and even shortterm soil moisture deficits can significantly reduce productivity. Therefore, soil moisture remains a robust and appropriate driver for assessing agricultural productivity.

Nonetheless, based on the reviewer #1 suggestion, we incorporated this analysis into the revised manuscript. Despite the acknowledged complexities, our conclusions are well supported by the model-simulated GPP response. Our response to reviewer #2 on the topic is repeated here:

"The aim of this study is to focus on hydroclimate metrics. Transpiration is the most dominant process contributing to AET on land and is strongly correlated with photosynthesis. Thus, an increase in AET serves as a reliable indicator of an increase in GPP. Consequently, we have chosen not to emphasize GPP in this analysis. However, we have revised portions of the manuscript to address this concern. Additionally, we included a plot illustrating the seasonal anomaly of GPP in the supplementary information (Figure S10) and provided a discussion on plant productivity in the conclusions section, along with examples of similar findings from other studies." (Conclusion section)

Regarding the selected drought indices, these are specifically designed to capture agricultural drought conditions and offer a comprehensive representation of soil moisture storage (SMDI across different soil depths) and key land—atmosphere interactions (ETDI), which reflect the balance between potential evapotranspiration and actual plant transpiration. These indices effectively capture both short- and long-term dry/wet conditions and avoid the known limitations of water balance models used for the PDSI (Narasimhan and Srinivasan, 2005).

As similar issues have be highlighted by earlier referee reports, but still not properly addressed by the authors, I recommend rejection.

The reviewer's comment is a little vague: "...similar issues...not properly addressed..." but without being explicit on which issues and how we failed to address them. This is the summary of the previous review round (major comments only):

- Reviewer #1 single major comment was related with the justification for the reference period. We addressed that comment by demonstrating in the revised manuscript that using other plausible reference periods the answers did not change qualitatively, and our conclusions remained the same regardless of which period was chosen as a reference (Additional figure (S2) is added in supplementary).
- Reviewer #2 had major comments but did not reject the paper: "I therefore recommend publication only after major revisions". Their major comments were: 1) a restructure of the paper to streamline the purpose and message more cleanly, which we had done; 2) discuss more about the model skill and in particular in the light of past ModelE studies, which we have done; 3) a question (not a major comment really) on why we did not use prescribed volcanic forcing and get more ensembles in, which we have answered; and 4) a request to add GPP or NPP anomaly plots, which we have added.

Then from this review round, which reviewer #5 had no opportunity to see before commenting, so only mentioning them here for completion):

- Reviewer #3 was positive with very few minor comments (answers above).
- Reviewer #4 had a number of comments (answers above), none of which were major, in our opinion. The only major point named by the reviewer was the lack of a statistical analysis of air pollution, which is not something we studied here in the first place, so the comment was not relevant.

In summary, we have substantially revised the manuscript to incorporate all suggestions from Reviewers 1 and 2, and we also included an analysis of the model-simulated GPP response, which supports and complements the conclusions drawn from the soil moisture-based drivers.

---

## Author Response (AR3)

**Dear editor,**

Thank you for your comments and decision. We have modified the abstract and conclusions as requested, by including all changes proposed. You can find the changes made in abstract and conclusion sections at the line number given below. Do let us know if more changes in the wording are needed.

**Abstract: L15-L25**

"Large volcanic eruptions can significantly influence the climate system by altering the stratospheric sulfate concentrations, atmospheric radiative balance, stratospheric and surface temperature, and regional hydroclimate. A comprehensive understanding of the volcanically-driven regional hydroclimate response is essential for assessing the socio-economic implications of such short-term episodic climate perturbations, which can lead to droughts, with potential significant impacts to vulnerable civilizations that depend on flooding as a natural crops irrigation mechanism. However, the regional rainfall response to volcanic forcings is substantially dominated by the natural variability in the rainfall response, and which creates a gap in our understanding of how eruptions affect ecohydrological conditions and plant productivity. "

**Conclusion: L682-L765**

"This study utilized the NASA GISS ModelE2.1 (MATRIX) Earth system modeling framework to investigate the mechanisms by which the 1991 Mt. Pinatubo eruption influenced hydroclimatic conditions and water-based drivers of plant productivity. This study successfully demonstrated the potential of volcanically-induced hydroclimate responses in context to complex but directly relevant impact metrics for societal implications, such as agricultural productivity. The application of process-based compound drought indices (SMDI and ETDI) targeting key climate-relevant quantities, including soil moisture, effectively address the challenge posed by the uncertainties associated with regional rainfall responses and presents a better assessment of its impact on plant productivity. Furthermore, the temporal as well as the latitudinal dependence of dominant hydroclimate drivers in shaping regional plant productivity response provides critical insights for understanding and guiding mitigation strategies to address the adverse impacts of such climate perturbation events."

**L 755-L761**

"Our findings demonstrate that soil moisture conditions across different regions can provide valuable insights into the full impacts on agricultural yields and regional carbon sink responses, particularly under scenarios involving implementation of solar geo-engineering (stratospheric

aerosol injection) or future large eruptions along with the changes in dynamic vegetation and crop cover. It needs to noted however that although in the future the physical mechanisms of a forcing response would be similar, the societal impact might be muted compared to that on past civilizations, due to present (and future) widespread irrigation practices and river level control in the presence of dams."

Regards,

Ram Singh, on behalf of all co-authors.